# Extracting interpretable signatures of whole-brain dynamics through systematic comparison

**Annie G. Bryant**[1]*, **Kevin Aquino**[1,2], **Linden Parkes**[3,4], **Alex Fornito**[4], **Ben D. Fulcher**[1]*

**1** School of Physics, The University of Sydney, Camperdown, New South Wales, Australia, **2** Brain Key Incorporated, San Francisco, California, United States of America, **3** Department of Psychiatry, Brain Health Institute, Rutgers University, Piscataway, New Jersey, United States of America, **4** School of Psychological Sciences, Turner Institute for Brain and Mental Health & Monash Biomedical Imaging, Monash University, Clayton, Victoria, Australia

* annie.bryant@sydney.edu.au (AGB); ben.fulcher@sydney.edu.au (BDF)

**Data Availability Statement:** All data used in this study is openly accessible, with preprocessed data provided in Zenodo at https://zenodo.org/records/10431855. UCLA CNP rs-fMRI imaging data can be

## Abstract

The brain's complex distributed dynamics are typically quantified using a limited set of manually selected statistical properties, leaving the possibility that alternative dynamical properties may outperform those reported for a given application. Here, we address this limitation by systematically comparing diverse, interpretable features of both intra-regional activity and inter-regional functional coupling from resting-state functional magnetic resonance imaging (rs-fMRI) data, demonstrating our method using case–control comparisons of four neuropsychiatric disorders. Our findings generally support the use of linear time-series analysis techniques for rs-fMRI case–control analyses, while also identifying new ways to quantify informative dynamical fMRI structures. While simple statistical representations of fMRI dynamics performed surprisingly well (e.g., properties within a single brain region), combining intra-regional properties with inter-regional coupling generally improved performance, underscoring the distributed, multifaceted changes to fMRI dynamics in neuropsychiatric disorders. The comprehensive, data-driven method introduced here enables systematic identification and interpretation of quantitative dynamical signatures of multivariate time-series data, with applicability beyond neuroimaging to diverse scientific problems involving complex time-varying systems.

## Author summary

Neuroimaging techniques, like resting-state functional magnetic resonance imaging (rs-fMRI), provide a window into complex brain dynamics in health and disease. Much existing work has involved manually distilling the complexity of a neural time-series dataset down to a set of hand-selected summary statistics, an approach that is prone to over-complicating or missing the most clearly interpretable and informative dynamical structures in the data. To overcome these methodological limitations, in this study, we introduce a systematic approach to capturing informative dynamical structure from neural time-series

downloaded from OpenNeuro with accession number ds000030 (https://openfmri.org/dataset/ds000030/). ABIDE rs-fMRI regional time series can be downloaded from Zenodo at https://zenodo.org/records/3625740. All code used to compute univariate and pairwise time-series features, perform classification tasks, and visualize results is provided on GitHub at https://github.com/DynamicsAndNeuralSystems/fMRI_FeaturesDisorders and Zenodo at https://zenodo.org/records/10467891.

**Funding:** L.P. was supported by the National Institute Of Mental Health of the National Institutes of Health under Award Number R00MH127296 (https://www.nimh.nih.gov/funding/training/nih-pathway-to-independence-award-k99-r00). The funder did not play any role in study design, data collection and analysis, decision to publish, or preparation in the manuscript.

data that compares across a broad range of interpretable analysis methods. Our framework encompasses five different representations with increasing complexity, from the localized activity of a single brain region up to the distributed activity of all brain regions and their pairwise interactions. We demonstrate our method in the context of four distinct neuropsychiatric disorders to discover and compare the types of brain activity dynamics that are most informative of each diagnostic group. We find that simpler techniques, like quantifying activity within a single brain region, perform surprisingly well in classifying schizophrenia and autism spectrum disorder cases from clinically normal controls—supporting continued investigations into region-specific alterations in neuropsychiatric disorders. Furthermore, combining region-specific metrics with inter-regional interactions generally provides a more informative understanding of how brain dynamics are altered in these conditions, demonstrating the benefit of combining local dynamics with pairwise coupling. Importantly, our systematic and comprehensive approach to quantifying interpretable patterns from complex time-series data shows promise for studying signatures of brain dynamics across domains, from functional fingerprinting to developmental trajectory analysis to dementia research.

## Introduction

Quantifying the dynamical structures in a complex system like the brain can clarify relationships across scales—for example, from individual neuronal populations to macroscopic brain states. In the brain, as with other complex systems, there are myriad ways to represent different types of dynamical patterns, many of which involve discretizing the brain into 'regions' [1, 2]. Neuroimaging modalities like functional magnetic resonance imaging (fMRI) measure brain activity within voxels that are typically aggregated into such brain regions, yielding a region-by-time multivariate time series (MTS). While an MTS can be analyzed using statistics derived from a vast theoretical literature on time-series analysis [3], only a limited set of methods are typically used to summarize fMRI data. For example, most resting state-fMRI (rs-fMRI) studies have examined functional connectivity (FC) between pairs of brain regions by computing the Pearson correlation coefficient [4, 5], a zero-lag cross-correlation that assumes a joint bivariate Gaussian distribution, which may not always hold with rs-fMRI data [5, 6].

An MTS comprised of rs-fMRI data can be summarized at multiple levels, including: (i) individual regional dynamics; (ii) coupling or communication between pairs of regions across distributed networks; and (iii) higher-order interactions amongst multiple regions [7]. The choice of how to represent the rich dynamical structure in a brain-imaging dataset—e.g., intra-regional activity (properties of the fMRI signal time series for a given region) or inter-regional coupling (statistical dependence of the fMRI signal time series for two regions)—and which statistical properties, or 'features', to measure is typically made manually by a given researcher. Of the few studies to directly compare alternative time-series features for functional connectivity, or (less commonly) intra-regional activity, the examined set of statistics is usually limited in scope and size [8–10]. Intra-regional activity is often overlooked in favor of inter-regional functional connectivity, in part because of the hypothesis that neuropsychiatric disorders arise from disruptions to inter-regional coupling and integration, with local dynamics alone insufficient to explain or predict diagnosis [11–16]. However, this hypothesis has yet to be evaluated systematically, and conclusions about the lack of localized disruptions are largely based on previous work that has examined regional properties based on graph theory (e.g., degree centrality or node strength for a seed region in an FC matrix) [17, 18] or a handful of

time-series features like the fractional amplitude of low-frequency fluctuations (fALFF) [19, 20] or regional homogeneity (ReHo) [21]—which could miss more nuanced changes to the blood oxygen-level dependent (BOLD) dynamics of a given brain region. Indeed, quantifying intra-regional activity can yield whole-brain maps of localized disruption, aiding interpretability and opening the door for new types of questions that are inaccessible using pairwise FC, such as how a given region responds to the targeted stimulation of another region [22].

Addressing the challenge of tailoring an appropriate methodology to a given data-driven problem, 'highly comparative' feature sets have recently been developed to unify a comprehensive range of algorithms from across the time-series literature, enabling broad and systematic comparison of time-series features (that would previously have been impractical). For example, the *hctsa* library includes implementations of thousands of interdisciplinary univariate time-series features [23, 24], and the *pyspi* library includes hundreds of statistics of pairwise interactions [25]. These algorithmic libraries have since been applied to systematically compare feature-based representations of diverse complex systems, from arm movement during various types of exercise [25] to light curves of different types of stars [26]. While it has been posited that rs-fMRI is too noisy and poorly sampled in time for more sophisticated time-series analysis methods (e.g., that have been applied to EEG data) [27–29], emerging applications of both *hctsa* and *pyspi* to rs-fMRI data have highlighted the utility of venturing beyond standard linear time-series analysis for both intra-regional [30] and inter-regional dynamics [31]. These highly comparative time-series feature approaches can distill the most informative biomarkers for a given application, such as distinguishing fMRI scans obtained at rest vs while watching a film [25] or comparing neurophysiological dynamics with cortical microarchitecture [32] or structural connectivity [33]. Importantly, time-series analysis methods developed outside of the neuroimaging space (such as dynamic time warping, originally developed for spoken word recognition [34]) have been shown to usefully quantify patterns in fMRI dynamics [35–37], underscoring the benefit of examining a comprehensive and interdisciplinary range of analysis methods.

To date, prior highly comparative analyses have focused on either intra-regional activity or inter-regional coupling separately, which fails to address which of the two—or indeed their combination—is optimal for a given problem. This question of which dynamical properties to capture from a complex dynamical system, and which algorithms might best capture them, is common to many data-driven studies, leaving open the possibility that alternative ways of quantifying dynamical structures may be more appropriate (e.g., yield better performance, clearer interpretation, or computational efficiency) for a given problem at hand. To address this gap, here we introduce a systematic method to evaluate multiple data-driven univariate time-series features, pairwise interaction statistics, and their combination, which is generally applicable to data-driven problems involving distributed time-varying systems. In the context of fMRI MTS, combining properties of intra-regional activity and inter-regional coupling is motivated by findings that the two representations synergistically improve classification performance across a variety of clinical settings including schizophrenia [11], Alzheimer's disease [38], nicotine addiction [39], and attention-deficit hyperactivity disorder [40]. Our method is applied to rs-fMRI for case–control classification of four neuropsychiatric disorders as an exemplary case study: schizophrenia (SCZ), bipolar I disorder (BP), attention-deficit hyperactivity disorder (ADHD), and autism spectrum disorder (ASD).

Neuroimaging-based diagnosis of psychiatric disorders is an area of great interest, as diagnosis is currently based on behavioral criteria [41] and is hindered by patient heterogeneity [42, 43] as well as inter-rater reliability issues [44, 45]. Increasingly large open-source rs-fMRI datasets spanning neuropsychiatric disorders like schizophrenia [46] and autism spectrum disorder [47, 48] have co-evolved with the publication of increasingly complex classifiers like

deep neural networks, which employ intricate algorithmic architectures that can be challenging to interpret [49–51]. Despite the improvements to in-sample classification performance often afforded by this increased complexity, core goals of neuroimaging-based classification studies include: (1) unpacking the neuroanatomical and functional underpinnings of a given disorder; and (2) developing a generalizable classification framework that aids the diagnosis of future (unseen) patients [41, 52–55]. Here, we implement linear support vector machine (SVM) classifiers to combine features based on clear algorithms derived from interdisciplinary literature, aiming to compare and contrast the types of underlying dynamical processes that distinguish brain activity across disorders—rather than absolute classification performance [22, 56]. Broadly, insights into the dynamical processes that are altered can deepen understanding of mechanisms underlying disease pathogenesis and progression, potentially yielding clinically translatable biomarkers [57, 58]. This study is accompanied by a repository that includes code to reproduce all presented analyses and visualizations [59].

## Materials and methods

An overview of our methodology is illustrated in Fig 1. We extracted the BOLD time series from each rs-fMRI volume, which were analyzed at the level of individual brain regions and pairs of brain regions (Fig 1A). We included participants from four neuropsychiatric disorder groups derived from two main cohort studies, as depicted in Fig 1B and summarized in Table 1. For each participant, we computed a set of time-series features reflecting the local dynamics of a single region's BOLD time series and coupling between a pair of brain regions (Fig 1C). We compared the performance of these univariate and pairwise features separately, and in combination, using linear classifiers for each disorder—specifically testing five different ways of capturing dynamical structure, as shown in Fig 1D and outlined in the following.

### Neuroimaging datasets and quality control

**UCLA CNP.**   Raw BOLD rs-fMRI volumes consisting of 152 frames acquired over 304 seconds were obtained from the University of California at Los Angeles (UCLA) Consortium for Neuropsychiatric Phenomics (CNP) LA5c Study [46]. This included cognitively healthy control participants as well as participants diagnosed with schizophrenia (SCZ), attention-deficit hyperactivity disorder (ADHD), and bipolar I disorder (BP). Details of diagnostic criteria and behavioral symptoms have been described previously, along with imaging acquisition details [46]. Imaging data were preprocessed using the *fmriprep* v1.1.1 software [62] and the independent component analysis-based automatic removal of motion artifacts (ICA–AROMA pipeline [63], as described previously [64]). Additional noise correction was performed to regress out the white matter, cerebrospinal fluid, and global gray-matter signals using the ICA–AROMA + 2P + GMR method [64–66]. Time series of 152 samples in length were extracted from the noise-corrected BOLD rs-fMRI volumes for 68 cortical regions [67] and 14 subcortical regions [68] spanning the left and right hemispheres.

**ABIDE.**   Preprocessed BOLD rs-fMRI time-series data consisting of between 49 and 433 time points (mean: 184 ± 63 time points) were obtained from the Autism Brain Imaging Data Exchange (ABIDE) I & II consortium [47, 48], working with a dataset specifically aggregated for an international classification challenge by [60] that included healthy controls and participants with autism spectrum disorder (ASD). Imaging data were preprocessed by the classification challenge authors [60, 69] using the standard pipeline from the '1000 Functional Connectomes Project', which includes motion correction, skull stripping, spatial smoothing, segmentation, and noise signal regression [69]. We opted to use the preprocessed time series corresponding to the Harvard–Oxford cortical atlas, which is published with the FSL software

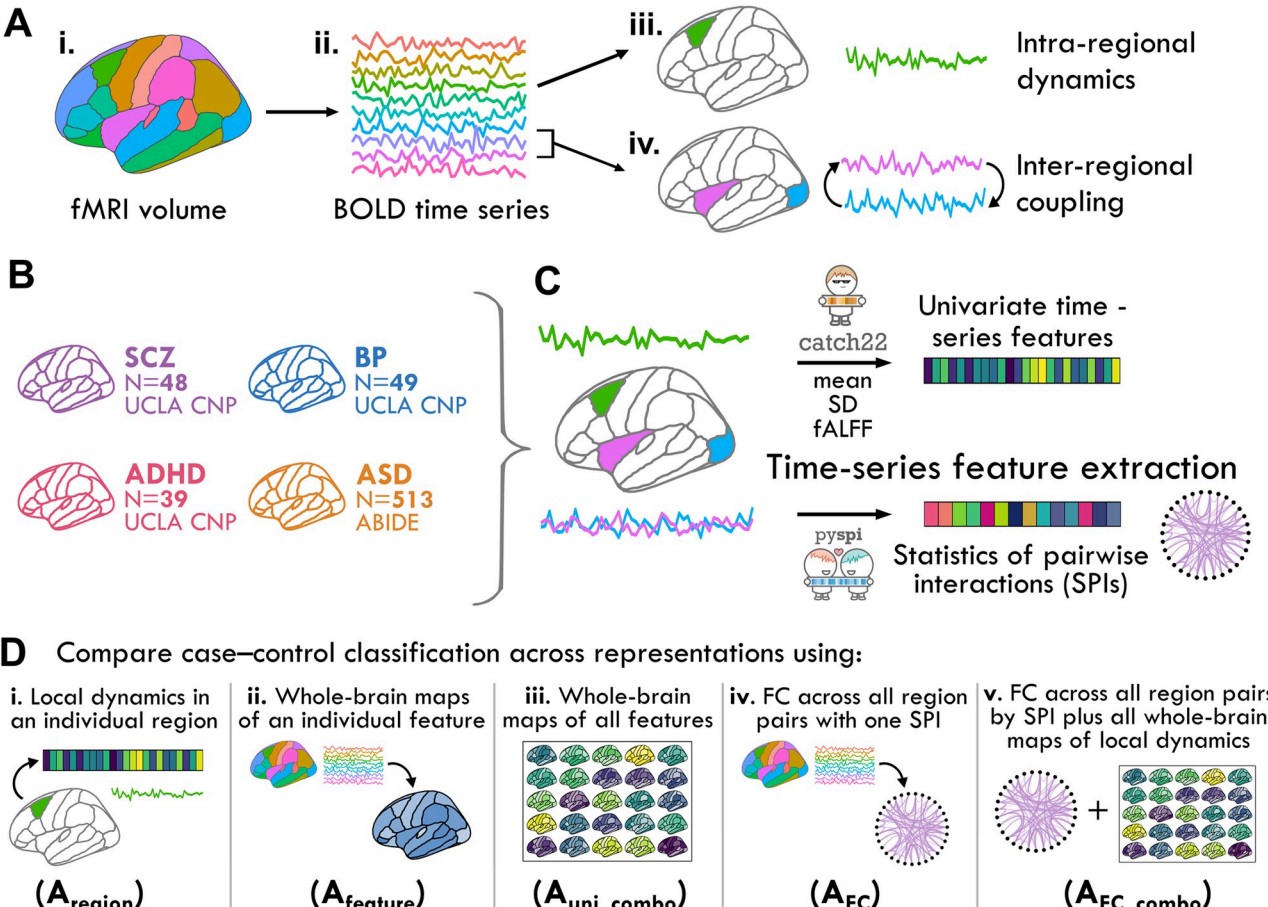

**Fig 1. In this study, we systematically compared different ways of quantifying dynamical patterns from resting-state fMRI time series, focusing on statistics of local regional dynamics and pairwise coupling across four neuropsychiatric disorders. A**. For a given resting-state fMRI volume (**i**), the cortex and subcortex are divided into individual regions from which the voxel-averaged BOLD signal time series is extracted (**ii**). Two key ways of quantifying dynamical patterns from these data are to: (**iii**) measure properties of the dynamics of individual brain regions (green); or (**iv**) to compute statistical dependencies between pairs of regions (pink and blue). **B**. To evaluate the performance of different types of dynamical representations of an fMRI time-series dataset (for identifying disease-relevant changes in neural activity), we included four neuropsychiatric exemplars originating from two open-access datasets: the UCLA CNP LA5c study [46] and the ABIDE I/II studies [47, 48, 60]. Each of the two cohorts also included cognitively healthy controls for comparison ($N = 116$ for UCLA CNP, $N = 578$ for ABIDE). **C**. For each type of dynamical structure that we extracted from an fMRI dataset (i.e., for each feature-based 'representation' of the data), we computed interpretable time-series features that encapsulate a diverse range of activity properties. Local dynamical properties were quantified from each brain region using a set of 25 time-series features (the *catch22* feature set [61] along with the mean, SD, and fALFF). Interactions between all pairs of regions were quantified using a set of 14 statistics of pairwise interactions (SPIs), comprising a representative subset from the *pyspi* package [25]. Values for both univariate time-series features and sets of pairwise coupling strengths from SPIs were captured in feature vectors that encapsulate a given type of interpretable dynamical structure(s). **D**. We evaluated case–control classification performance for each neuropsychiatric disorder using linear SVM classifiers fit to each of five distinct ways of representing resting-state fMRI properties: (i) all 25 univariate time-series features in an individual region, $A_{region}$; (ii) whole-brain maps of an individual time-series feature, $A_{feature}$; (iii) whole-brain maps of all 25 univariate time-series features, $A_{uni\_combo}$; (iv) functional connectivity (FC) networks across all region pairs using one SPI, $A_{FC}$; and (v) $A_{FC}$ along with all 25 univariate time-series features computed from all brain regions ($A_{uni\_combo}$), termed $A_{FC\_combo}$.

library [70] and includes 48 regions, noting that homotopic region pairs were pre-consolidated into singular bilateral regions.

**Quality control.** After preprocessing, we identified six participants from the UCLA CNP dataset (IDs: sub-10227, sub-10235, sub-10460, sub-50004, sub-50077, and sub-70020) for whom the ICA–AROMA + 2P + GMR pipeline yielded constant zero values across all time-points and brain regions. These six participants ($N = 3$ Control, $N = 2$ SCZ, $N = 1$ ADHD) were excluded from all further analyses. Head movement in the scanner was evaluated using

**Table 1. Key demographics for participants included in this study after quality control exclusion processes.**

| Study | Diagnosis | N | % Female (N) | Age; Mean (SD) |
|---|---|---|---|---|
| ABIDE | ASD | 513 | 14.6 (75) | 17.5 (10.5) |
| | Control | 578 | 26.0 (152) | 17.1 (9.5) |
| UCLA CNP | ADHD | 39 | 48.7 (19) | 31.6 (10.1) |
| | BP | 49 | 42.9 (21) | 35.3 (9.0) |
| | SCZ | 48 | 25.0 (12) | 36.6 (9.0) |
| | Control | 116 | 46.6 (54) | 31.2 (8.7) |

framewise displacement (FD) calculated from the six framewise head-motion parameters (corresponding to rotation and translation in the *x*, *y*, and *z* planes) using the method of [71]. The mean FD was compared between each disorder and the corresponding control group using a Wilcoxon rank-sum test, implemented in R with the `wilcox_test` function from *rstatix* (version 0.7.2) We found that SCZ, BP, and ASD—but not ADHD—cases exhibited significantly higher mean FD relative to control participants (all $P < 0.05$), consistent with prior studies [72–74]. In order to mitigate the potential confounding effect of head motion on the dynamical BOLD properties with putative neural causes, we applied the movement-based exclusion criteria described in [65] (labeled as 'lenient' criterion in that study), such that any participant with mean FD >0.55 mm was excluded from further analysis. This step excluded two participants from the UCLA CNP study ($N = 1$ SCZ, $N = 1$ Control) and 59 participants from the ABIDE study ($N = 36$ ASD, $N = 23$ Control).

**Participant summary statistics.** After performing quality control, we retained between $N = 39$ and $N = 578$ participants per clinical group across the two studies, with summary statistics provided in Table 1 and demographic distributions visualized in S3 Fig. Participants in the UCLA CNP dataset were scanned at two different sites [46] and participants in the ABIDE dataset were scanned across seventeen sites [69]. We did not include site information as a covariate in our classifiers, nor did we perform cross-site alignment, to simply focus on how our method performs in a larger heterogeneous clinical sample. As a robustness analysis, we did examine the performance of $A_{region}$, $A_{feaure}$, and $A_{uni\_combo}$ within the two largest ABIDE sites individually (Sites #5 and #20) to compare classification performance within versus across sites.

## Time-series feature extraction

**Univariate features.** For a given brain region, the BOLD time series can be summarized using a set of univariate time-series features. We opted to use the *catch22* library of univariate properties [61], which are listed in Table 2. The *catch22* features were drawn from the broader *hctsa* [23, 24] library, designed to be a highly explanatory subset of time-series features for generic time-series data mining problems [75] (though not defined specifically for neuroimaging datasets). To compare the performance of these features with a standard biomarker for quantifying localized resting-state BOLD activity, we also computed the fractional amplitude of low-frequency fluctuations (fALFF), which measures the ratio of spectral power in the 0.01–0.08 Hz range to that of the full frequency range [19]. fALFF is considered to be an index of spontaneous activity in individual brain regions and is fairly robust to the noise, scanner differences, and low sampling rates typical of rs-fMRI data [20, 76, 77]. The *catch22* features were computed in R using the 'theft' package (version 0.4.2; [78]) using the `calculate_features` function with the arguments `feature_set = "catch22"`, which *z*-scores each time series prior to feature calculation. We also calculated the mean and standard

**Table 2. The 25 univariate time-series features computed for each brain region to capture intra-regional dynamics.** For each univariate time-series feature (shown in the 'Feature name' column), a brief description is given for the property the corresponding feature captures.

| Feature name | Feature description |
| --- | --- |
| ACF_first_min | First minimum of the autocorrelation function (ACF) |
| ACF_timescale | First 1/e crossing of the ACF |
| AMI_timescale | First minimum of the automutual information function (linear version) |
| AMI2 | Histogram-based automutual information (lag 2; 5 bins) |
| centroid_freq | Frequency corresponding to the centroid of the power spectral density |
| DFA | Detrended fluctuation analysis (low-scale scaling) |
| embedding_dist | Goodness of exponential fit to embedding distance distribution |
| entropy_pairs | Entropy of successive pairs of values in symbol-discretized time series |
| fALFF | Ratio of spectral power in the 0.01–0.08Hz range to the full frequency range |
| forecast_error | Error of 3-point rolling mean forecast |
| high_fluctuation | Proportion of high incremental changes in the series |
| low_freq_power | Power in the lowest 20% of frequencies |
| mean | Mean of the time series values |
| mode_10 | Mode of a 10-bin histogram of time-series values |
| mode_5 | Mode of a 5-bin histogram of time-series values |
| outlier_timing_neg | Negative outlier timing (higher = later in time series) |
| outlier_timing_pos | Positive outlier timing (higher = later in time series) |
| periodicity | First local maximum of the ACF following Wang et al. [79] criteria |
| rs_range | Rescaled range fluctuation analysis (low-scale scaling) |
| SD | Standard deviation of time series |
| stretch_decreasing | Longest stretch of decreasing values |
| stretch_high | Longest stretch of above-mean values |
| transition_variance | Transition matrix column variance |
| trev | Time reversibility (nonlinear autocorrelation measure) |
| whiten_timescale | Change in autocorrelation timescale after incremental differencing |

deviation (SD) from the raw BOLD time series by including the `catch24 = TRUE` argument, which yielded a total of 24 univariate time-series features. We computed fALFF in Matlab as per the code implementation included in [33]. Collectively—the *catch22* features, mean, SD, and fALFF—comprise 25 univariate features that encapsulate a diverse set of statistical time-series properties with which to compare across different clinical exemplars. These 25 time-series features were computed across all brain regions and participants, yielding a three-dimensional array in the form of $N \times R \times F$—comprised of $N$ participants, $R$ brain regions, and $F$ time-series features.

**Pairwise features.** To summarize different types of pairwise coupling, or functional connectivity (FC), across pairs of brain regions, we used a subset of 14 SPIs from the *pyspi* library of pairwise interaction statistics [25]. Each of these SPIs is listed along with its code name in 'pyspi' in Table 3, along with information about the literature category and directionality per SPI. These 14 SPIs contain a single representative from each of the 14 data-driven modules of similarly performing SPIs described in [25]. It includes the most commonly used FC metric for fMRI time series, Pearson correlation coefficient (denoted as 'cov_EmpiricalCovariance' in *pyspi*), enabling direct comparison of its performance to the other thirteen evaluated SPIs, which span nonlinear coupling (e.g., additive noise modeling), frequency-based coupling (e.g., power envelope correlation), asynchronous coupling (e.g., coherence magnitude), time-lagged coupling (e.g., dynamic time warping), and directed coupling (e.g., directed information). The

**Table 3. The 14 statistics for pairwise interactions (SPIs) computed across brain region pairs.** For each statistic of pairwise interactions (SPI; shown in the 'SPI' column), a brief description is given for the type of interaction that the corresponding SPI captures.

| SPI | Description |
| --- | --- |
| PSI_frequency | Phase slope index (PSI) of the frequency domain, measures information flow based on complex-valued coherency |
| ANM | Additive noise model, tests for directed nonlinear dependence of x $\to$ y |
| DI | Directed information, measures information flow from one time series to another without any time-lags |
| transfer_entropy | Transfer entropy, measures information flow from one time series to another with Takens time-delay embedding |
| phi_star | Phi-star, integrated information proxy that measures information lost when two time series are disconnected |
| spectral_GC | Spectral Granger causality, frequency domain equivalent to Granger causality (nonparametric) |
| PLI | Phase lag index, measures phase synchronization across the full frequency range |
| PSI_time_frequency | PSI of the time-frequency domain, measures information flow based on complex-valued coherency |
| barycenter_DTW | Barycenter, the univariate time series that captures the center of mass between two time series |
| DTW | Dynamic time warping, measures the minimum distance between two (potentially dilated) time series |
| power_envelope_corr | Power envelope correlation, measures correlation between power envelopes of two time series |
| coherence_magnitude | Coherence magnitude, measures mean coherence between two time series for full frequency range |
| cointegration | Cointegration, tests if the linear combination of two time-series has a lower integration order |
| Pearson | Pearson correlation coefficient—computed as empirical covariance, which is equivalent as we z-score data |

14 SPIs were computed from the BOLD time series for each pair of brain regions using the *pyspi* package (version 0.4.0) in Python [80]. Given the large number of MTS matrices to process (across all participants), we distributed *pyspi* computations on a high-performance computing cluster. Each SPI yielded a matrix of pairwise interactions between all brain regions, with a total of 6642 possible pairs for UCLA CNP (from $R = 82$ brain regions) and 2256 pairs for ABIDE (from $R = 48$ brain regions), after omitting self-connections. As some SPIs (like the Pearson correlation) are undirected, the resulting FC matrix is symmetric, and only the set of coupling values for all unique brain-region pairs (corresponding to the upper half of the FC matrix) is retained for these SPIs. Across all SPIs, *pyspi* computations yielded a three-dimensional array in the form of $N \times P \times S$, comprised of $N$ participants, $P$ region pairs (noting that this value will differ for directed versus undirected SPIs), and $S$ SPIs.

## Case–control classification

We sought to comprehensively compare the ability of diverse intra-regional time-series features and inter-regional coupling strengths to separate healthy controls from individuals with a given clinical diagnosis. Univariate, intra-regional time-series features are sensitive to local, region-specific disruptions in neural activity, whereas inter-regional SPIs are sensitive to disrupted coupling between pairs of brain regions. We first asked whether case versus control separation would be better with: (i) multiple dynamical properties measured from a single brain region; (ii) a single time-series property measured across brain regions; or (iii) the

combination (the full set of time-series properties across all brain regions). We then asked how pairwise FC measurements could capture case–control differences, and whether including our set of diverse univariate features would enhance SPI-wise classification performance. To compare the ability of univariate and/or pairwise features to separate case versus control individuals, we evaluated five different classification pipelines as enumerated in the following subsections, labeling each analysis as 'A$_X$' where X captures a short description of the analysis. These are listed as follows:

1. A$_{region}$: The performance of individual brain regions given all $F$ univariate time-series features, assessed via classifiers fit to $N \times F$ (participant × feature) matrices as depicted in Fig 1D(i). $R$ = 82 models are fit with this representation for SCZ, BP, and ADHD, and 48 models for ASD.

2. A$_{feature}$: The performance of individual univariate time-series features measured from all brain regions, assessed via classifiers fit to $N \times R$ (participant × region) matrices as depicted in Fig 1D(ii). $F$ = 25 such models are fit for all four disorders.

3. A$_{uni\_combo}$: The performance of all $F$ univariate time-series features computed across all $R$ brain regions, assessed via classifiers fit to $N \times RF$ (participant × brain-region—feature) matrices as depicted in Fig 1D(iii). Only one such model is fit per disorder.

4. A$_{FC}$: The performance of the set of coupling strengths measured from all brain region pairs using an individual SPI, assessed via classifiers fit to $N \times P$ (participant × region–pair) matrices, as depicted in Fig 1D(iv). $S$ = 14 such models are fit for each disorder.

5. A$_{FC\_combo}$: The performance of an individual SPI measured from all brain region pairs with the addition of all $F$ univariate time-series features, assessed via classifiers fit to $N \times PRF$ (participant × region-pair–region–univariate-feature) matrices as depicted in Fig 1D(v). $S$ = 14 such models are fit for each disorder.

**Classifier fitting and evaluation.**   Each of the analyses enumerated in the previous sections (e.g., A$_{region}$) refers to a way in which we extracted a set of interpretable time-series features from an fMRI dataset to capture different types of dynamical statistics. We sought to compare how these feature-based representations could distinguish cases from controls in each neuropsychiatric disorder using a simple measure of classification performance. Classification was evaluated using linear SVMs [81], which comprise a simple classical machine learning model that can handle large input feature spaces using regularization [82].

The UCLA CNP cohort is characterized by class imbalances (such that there are more control participants than those with a given neuropsychiatric disorder), so we applied inverse probability weighting, which weights each sample according to the inverse frequency of its corresponding class to increase the impact of the minority class on the decision boundary [83, 84]. Additionally, we evaluated classifier performance using balanced accuracy [85], which measures the average between classifier sensitivity and specificity to reflect accuracy in both the majority and minority classes, as recommended for neuroimaging data with class imbalances [84].

Classifiers were evaluated using stratified 10-fold cross-validation (CV) such that the hyperplane was defined based on the training data and used to predict the diagnostic labels of the unseen test data for each fold. In order to reduce outlier-driven skews in time-series feature distributions, each training fold was normalized using the scaled outlier-robust sigmoidal transformation described in [24] and the same normalization was applied to each test fold to prevent data leakage [86]. We confirmed that this reduced more distributional skew of many

univariate and pairwise time-series features compared to *z*-score normalization, as shown in S15 Fig. To ensure that results were not driven by any one specific CV split, each 10-fold CV procedure was repeated 10 times [87–89], yielding 100 out-of-sample balanced accuracy metrics per model. We report the mean ± SD tabulated across the 100 out-of-sample balanced accuracy metrics per model. As a robustness analysis, we subsequently compared the performance of each model (e.g., the left pericalcarine cortex in $A_{region}$) with the same 10-repeat 10-fold cross-validation as in the main analysis using each of: a linear SVM with L1 ('LASSO') coefficient regularization [90] and squared-hinge loss, an SVM with a radial basis kernel, a random forest ensemble, or a gradient boosted decision tree ensemble. The standard linear SVM with hinge loss and without L1 coefficient regularization yielded the highest classification performance overall (cf. S7(A) Fig), and as this linear SVM represented the most parsimonious and interpretable of the evaluated models, we focus on classification metrics from this model throughout the paper. We also evaluated the effect of nested cross-validation to tune the regularization parameter, *C*, across values ranging between [0.0001, 100]—as well as the inclusion (or exclusion) of inverse probability weighting—finding that this also did not generally improve performance beyond our *a priori* parameter settings of *C* = 1 and applying inverse probability weighting (cf. S7(B) Fig and S16 Table).

The main linear SVM classifiers were implemented in Python with the `SVC` function from *scikit-learn* [91]) with a default value of 1 for the regularization parameter, *C*, and with the `class_weight = 'balanced'` argument for inverse probability weighting. The scaled outlier-robust sigmoidal transformation was applied using a custom Python script adapted from the Matlab-based `mixedSigmoid` normalization function from *hctsa* [23, 24]. We implemented 10 repeats of 10-fold CV using the `RepeatedStratifiedKFold` function from *scikit-learn* as part of a 'pipeline', which also automatically applied normalization based on training fold parameters and evaluated balanced accuracy for each test fold. We set the random number generator state for the `RepeatedStratifiedKFold` function such that all compared models were evaluated on the same resampled sets of test folds (for a total of 100 test folds). While the balanced accuracy is only computed for the default decision threshold for each classifier, we also computed the area under the receiving operator characteristic curve (AUC) for each test fold to evaluate model performance across thresholds. Models achieving greater than 50% balanced accuracy (i.e., better than chance) showed similar performance between balanced accuracy and AUC (Pearson's *R* between 0.77–0.97), so we focus on balanced accuracy in this work to align with the binary predictions of diagnostic group [84]. The other classifiers in our robustness comparison were also implemented with *scikit-learn*, using the `RandomForest`, `SVC`, `RandomForest`, or `GradientBoostingClassifier` functions, and nested cross-validation was performed with the `RandomizedSearchCV` function.

**Dimensionality reduction and feature selection.**   While $A_{region}$ and $A_{feature}$ both entail models with more samples than features, the other three representations—$A_{uni\_combo}$, $A_{FC}$, and $A_{FC\_combo}$—involve models with more features than samples, which risks overfitting to accumulated noise [92]. To test for such overfitting, we evaluated the change in performance after reducing the SVM feature space with principal components analysis (PCA), focusing on analysis $A_{uni\_combo}$ as a starting point. PCA was implemented using the `PCA` function from the *FactoMineR* package in R (version 2.9) [93]. Since classification models based on individual brain regions yielded the lowest-dimensional classifiers ($A_{region}$: 25 time-series features per model), we selected the first 25 principal components (PCs) from the region × feature PCA for each case–control comparison group for consistency. Scores for these 25 PCs are provided in S17 Table, and the first two PCs are plotted for each disorder in S6(A) Fig. Each case–control comparison was evaluated using these 25 PCs with the same repeated 10-fold classification procedure as described above.

To fairly compare model performance across representations, we also performed data-driven feature selection for all representations except for $A_{uni\_combo}$. For $A_{region}$ and $A_{feature}$—the two representations where $p < n$—we implemented a strategy in which for each of the 100 folds (10 folds resampled 10 times), we identified the brain region (for $A_{region}$) or time-series feature (for $A_{feature}$) that yielded the best in-sample training fold balanced accuracy, and retained its corresponding out-of-sample test balanced accuracy. For example, for $A_{region}$ in SCZ, in fold #1 out of 100, the left pericalcarine might yield the highest in-sample balanced accuracy, so we would retain the out-of-sample balanced accuracy for the left pericalcarine for this fold. For fold #2, the right cuneus might yield the highest in-sample balanced accuracy, so we would retain the right cuneus out-of-sample balanced accuracy, and so on for the rest of the folds. In the event of a tie for a given fold, we computed the mean out-of-sample balanced accuracy between the tied brain regions. As only one model was evaluated for $A_{uni\_combo}$, we simply examined the distribution of 100 out-of-sample balanced accuracy values.

By contrast, $A_{FC}$ and $A_{FC\_combo}$ both involve $p \gg n$ feature matrices that increase the potential for overfitting, so multiple SPIs yielded 100% in-sample balanced accuracy for each training fold— precluding the utility of this method for selecting one representative property of inter-regional coupling per fold. As an alternative selection criterion, we fit a PCA for each SPI (or SPI with local univariate properties for $A_{FC\_combo}$) and retained the first 10 PCs. Within each training fold, we supplied these 10 PCs as input feature matrices to the same linear SVM classifier as above, in order to identify the SPI which (in PC-space) yielded the highest in-sample training balanced accuracy. Upon identifying that top-performing SPI, we used its full feature-space matrix as the input to a linear SVM classifier to measure its out-of-sample balanced accuracy for the corresponding fold. In the event of a tie for a given fold, we computed the mean out-of-sample balanced accuracy between the tied SPIs (assessed in the full feature space). This selection method provided a heuristic for selecting the top SPI in a lower-dimensional space while preserving input information for the out-of-sample classification metrics we evaluated across representations.

**Statistical significance.** To make statistical inferences about how a given classifier performed relative to chance level, we calculated the probability of observing a given mean balanced accuracy relative to a null distribution. For each model, we fit 1000 SVM classifiers with randomly shuffled diagnostic class labels using the same 10-repeat 10-fold CV as described above—yielding 1000 null balanced accuracy estimates per model. After confirming that all evaluated models yielded approximately normal null distributions centered at 50% (i.e., chance or predicting all the same class) balanced accuracy (S16 Fig), we derived the mean and standard deviation from each null distribution. Instead of fitting a full non-parametric permutation test for each specific model (with e.g. 10, 000 or more permutations), we instead imposed a Gaussian distribution with two moments (i.e., mean and SD) to approximate the cumulative distribution function per model. Each observed balanced accuracy metric was compared with its corresponding cumulative distribution function using the `pnorm` function in R to obtain a one-tailed $P$-value, capturing the probability of obtaining a null balanced accuracy greater than or equal to the observed balanced accuracy. The cumulative distribution function step considerably reduced computational expenses, enabling the measurement of very small $P$-values that would be computationally prohibitive to detect in a full nonparametric treatment considering the number of models tested throughout this study. In order to correct for multiple comparisons within each disorder, we applied Benjamini–Hochberg correction [94] to control the false discovery rate at the $\alpha = 0.05$ level. All Benjamini–Hochberg corrected $P$-values are denoted as $P_{corr}$ (unless otherwise specified) and the number of comparisons are indicated as appropriate.

**Corrected $T$-statistics.** To quantify the change in classifier performance for pairwise FC matrices with versus without the inclusion of univariate region × feature data (from

A$_{uni\_combo}$), we compared the balanced accuracy distributions across all 100 test folds for the corresponding two classifier inputs per SPI. The standard $T$-test for group means is suboptimal in this case, as the participant overlap across test folds in the two compared models violates the assumption of independent samples [95]. We instead implemented a corrected two-tailed test statistic $T_{corr}$ designed specifically for repeated $k$-fold cross-validation, as defined in [96] based on the original corrected T-statistic described in [97]. This correction was applied using the `repkfold_ttest` function from *correctR* package in R (version 0.2.1) [98].

**Confound analysis based on age, sex, and head motion.** The proportion of males versus females was compared between each disorder and the corresponding control group using a $\chi^2$ test, implemented in R with the `chisq_test` function from *rstatix* (version 0.7.2). We compared age and mean framewise displacement (FD) distributions for each disorder and the corresponding control groups using Wilcoxon rank-sum tests, implemented in R with the `wilcox_test` function from *rstatix* (version 0.7.2). To evaluate how well these confound variables could predict clinical diagnosis on their own, we fit the same 10-repeat 10-fold linear SVM classification pipeline to sex data (with male encoded as 0 and female encoded as 1), age data, and mean FD data. Results from this robustness analysis are depicted in S3(D) Fig. We used these classification metrics as a baseline from which we could assess the improvement in case–control distinction using BOLD-derived time-series features. In addition to the performance of these confound variables on their own and that of all the models we evaluated through our main analysis, we additionally evaluated the classification performance of models trained on BOLD time-series features plus confound variables as a robustness analysis. These classification metrics are shown side-by-side in S3(E) Fig.

**Volumetric analysis of group differences.** In order to test whether the balanced accuracy of a given brain region showed a linear relationship with gray matter volume differences in case–control comparisons, we compared the number of voxels in each brain region per participant in the UCLA CNP cohort (as volumetric data from the Harvard–Oxford cortical atlas is not available for the ABIDE ASD cohort). We used the *fmriprep* anatomical derivative 'aparc +aseg' parcellation and segmentation volumes that were mapped to MNI152 space using nonlinear alignment and tabulated the number of voxels in each region. Of note, these segmentation volumes correspond to the Desikan–Killiany–Tourville atlas [99] in which voxels for three brain regions (frontal pole, temporal pole, and banks of the superior temporal sulcus) were redistributed into other regions—thus precluding volumetric analyses for those three regions. For each brain region, we regressed mean volume (measured in the number of voxels) on diagnosis for each case–control comparison using an ordinary least squares model, with the resulting $\beta$ coefficients listed in Supplementary S2 Table. We examined the magnitude of the $\beta$ coefficient estimated for each brain region to evaluate whether volumetric differences relative to controls were related to that region's balanced accuracy. The Pearson correlation coefficient was computed to measure the linear association between $\beta$ coefficient magnitude and classification performance for all 79 brain regions per disorder. Additionally, to evaluate whether the overall volume of a region was related to how well its resting-state activity distinguished cases from controls, we compared the mean number of voxels per region across participants per disorder with the disorder-wise mean balanced accuracy per region using the Pearson correlation coefficient.

## Data visualization

All figures in this study were compiled using R (v. `4.3.2`) using the *ggplot2* package (v. `3.5.1`; [100]) unless otherwise specified. Brain maps were visualized using the *ggseg* package (v. `1.6.5`; [101]) along with the *ggsegHO* package (v. `1.0.2.001`) for the Harvard-Oxford cortical

atlas. Raincloud plots were generated with the `geom_violinhalf` function from the *see* package (v. `0.8.3`; [102]). Heatmaps with dendrograms were created using the `ComplexHeatmap` function from the *ComplexHeatmap* package (v. `3.18`, [103, 104]). Venn diagrams were created using the `venn.diagram` from the *VennDiagram* package (v. `1.7.3`; [105]).

## Results

Starting with an rs-fMRI blood oxygen level-dependent (BOLD) signal MTS, in the form of brain regions sampled over time as shown in Fig 1A(i) and 1A(ii), we quantified both intra-regional dynamics (Fig 1A[iii]) and pairwise coupling (Fig 1A[iv]) using interpretable time-series features derived from a rich interdisciplinary scientific literature. We employed a case study comprising four neuropsychiatric disorders across two cohorts (as schematically depicted in Fig 1B and summarized in Table 1). To represent intra-regional BOLD activity fluctuations, we computed 25 univariate time-series features that included the *catch22* feature set [61], as schematically depicted in Fig 1C. The *catch22* features were distilled from an initial library of over 7000 candidate features [24] to concisely capture diverse properties of local dynamics, such as distributional shape, linear and nonlinear autocorrelation, and fluctuation analysis [61] (as described further in Methods Sec. 'Time-series feature extraction'). We also included three additional features: the mean and standard deviation (SD)—based on our previous findings that these first two moments of a time series distribution can be highly informative in a given application [106]—and fALFF as a benchmark statistic for localized rs-fMRI dynamics [19, 107]. Inter-regional FC was summarized using 14 statistics for pairwise interactions (SPIs) derived as a representative subset from an initial library of over 200 candidate SPIs in the *pyspi* library [25], including the Pearson correlation coefficient. This set of SPIs includes statistics derived from causal inference, information theory, and spectral methods, which collectively measure a variety of coupling patterns (directed vs. undirected, linear vs nonlinear, synchronous vs lagged) that paint a fuller picture of inter-regional communication patterns [8, 25, 35]. The full list of univariate and pairwise time-series features and brief descriptions are provided in Tables 2 and 3.

In order to understand how well localized intra-regional dynamical properties and/or inter-regional FC could distinguish cases from controls, we systematically compared five distinct representations of BOLD activity dynamics, labeled $A_{region}$, $A_{feature}$, $A_{uni\_combo}$, $A_{FC}$, and $A_{FC\_combo}$, as illustrated in Fig 1D(i)–1D(v):

1. $A_{region}$ (Fig 1D[i]) represents each rs-fMRI time series using a set of features capturing 25 dynamical properties of an individual brain region. Strong classification performance in this representation suggests localized, region-specific changes in neural activity in the corresponding disorder.

2. $A_{feature}$ (Fig 1D[ii]) represents each rs-fMRI time series using a given time-series feature that captures a single dynamical property from all brain regions (82 for SCZ, BP, and ADHD, and 48 for ASD). Strong classification performance in this representation suggests that a specific property of intra-regional resting activity is altered in one or more part(s) of the brain in the corresponding disorder.

3. $A_{uni\_combo}$ (Fig 1D[iii]) represents each rs-fMRI time series using a set of features combining the 25 time-series properties evaluated across all brain regions. For this representation to out-perform $A_{region}$ and/or $A_{feature}$ individually, it suggests changes to different time-series properties across different regions that could not be fully captured by either reduced representation.

4. $A_{FC}$ (Fig 1D[iv]) represents each rs-fMRI time series using a set of features that capture all pairs of inter-regional coupling strengths computed using a single SPI. Strong classification performance in this representation suggests that a specific type of inter-regional coupling is altered amongst one or more brain region pairs in the corresponding disorder.

5. $A_{FC\_combo}$ (Fig 1D[v]) represents each rs-fMRI dataset as a set of features that capture all pairs of inter-regional coupling from a given SPI (as $A_{FC}$) as well as 25 time-series properties of all individual brain regions (as $A_{uni\_combo}$). For this representation to outperform $A_{FC}$, it suggests the presence of complementary and disease-relevant information in both localized intra-regional activity and pairwise coupling that is not captured with FC alone.

We first discuss each analytical representation individually to examine the types of biological and methodological insights each representation affords, then we systematically compare the performance of cross-validated classifiers based on each of the five representations. In all analyses, classification performance was evaluated using cross-validated balanced accuracy (i.e., the arithmetic mean of sensitivity and specificity, cf. [85]) measured using linear SVM classifiers; see Methods Sec. 'Case–control classification' for details on classifier selection.

## Analysis $A_{region}$: Characterizing localized changes to intra-regional dynamics across disorders

With Analysis $A_{region}$, we propose that if the BOLD activity in a single brain region is disrupted across patients relative to controls, the intra-regional dynamical properties of that region can meaningfully distinguish cases from controls in the corresponding disorder (as depicted in Fig 1D[i]). Such a result would contrast with previous studies positing that neuropsychiatric disorders like schizophrenia and ADHD are characterized by inter-regional dysfunction across distributed networks, rather than spatially localized disruptions [14, 107–109]. By reducing the complexity of an individual's rs-fMRI data down to 25 time-series features that encapsulate the dynamical properties of an individual region, we can interpret the results through a spatial map of disrupted fMRI dynamics in each disorder. As shown in Fig 2A, multiple brain regions exhibited dynamics that significantly distinguished cases from controls (defined as Benjamini–Hochberg corrected $P_{corr} < 0.05$, one-tailed permutation test) in SCZ (34/82 individual regions) and in BP (4/82 regions), but not in ADHD or ASD; see S1 Table for full brain region balanced accuracy results. The spatial distribution of intra-regional differences in underlying dynamics (assessed via cross-validated balanced accuracy) is shown for each brain region in Fig 2B(i) and 2B(ii) for SCZ and BP, respectively; results for ADHD and ASD are shown in S1 (C) Fig, though we focus on SCZ and BP here.

In SCZ, BOLD dynamics of medial occipital brain regions were the most informative of diagnosis, as shown in Fig 2B(i). Specifically, against a background of half of all brain regions yielding significant classification performance, dynamical signatures of activity in the left pericalcarine cortex best classified SCZ cases from controls, with 72.7 ± 11.2% (mean ± standard deviation) balanced accuracy (one-tailed permutation test, $P_{corr} = 2 \times 10^{-4}$, corrected across 82 brain regions), followed closely by the right cuneus (69.3 ± 12.6%, $P_{corr} = 2 \times 10^{-3}$) and left cuneus (68.0 ± 11.0%, $P_{corr} = 3 \times 10^{-3}$; see S1(A) Fig for individual region test-fold distributions). The relatively high performance of these medial occipital structures was unique to SCZ (compared to the other three disorders), which may reflect an SCZ-specific signature of localized alterations to visual areas. By contrast, subcortical structures like the right hippocampus (67.4 ± 14.4%, $P_{corr} = 0.02$) and left thalamus (65.1 ± 11.9%, $P_{corr} = 0.03$) distinguished BP cases from controls with the highest balanced accuracy. Among the significant brain regions

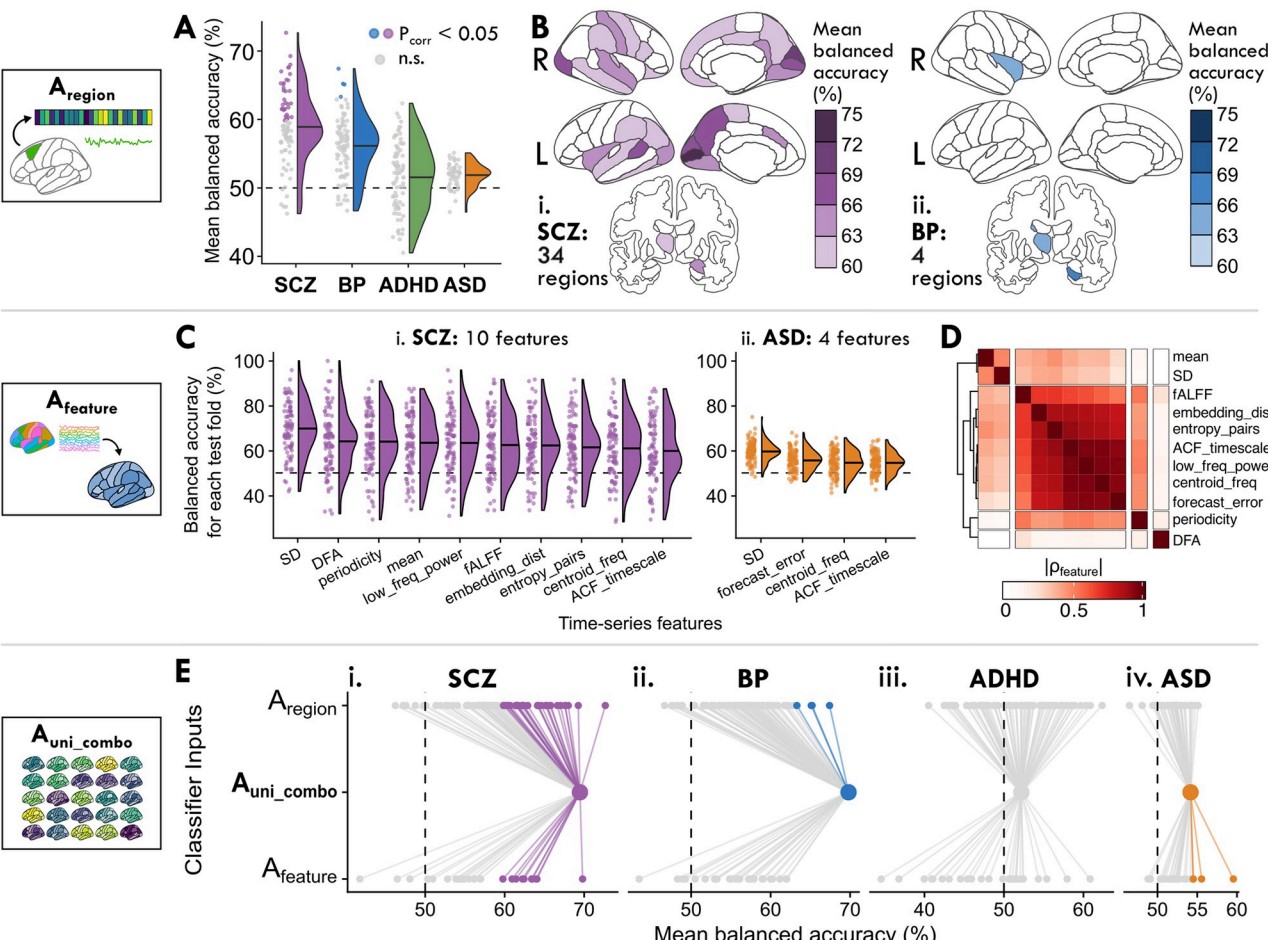

**Fig 2. Local properties of BOLD dynamics within individual regions distinguish SCZ and BP cases from controls, and classification performance improves by combining all regions and time-series features. A**. We investigate how the dynamics of a given individual brain region can distinguish cases from controls per disorder in $A_{region}$ (cf. Fig 1D[i]). The balanced accuracy is shown for all brain regions as a raincloud distribution, where each point corresponds to the mean classification performance (over 100 cross-validated test folds) of a single region. Points are colored to indicate a statistically significant balanced accuracy metric (Benjamini–Hochberg $P_{corr} < 0.05$, corrected across 82 brain regions for SCZ, BP, and ADHD and 48 brain regions for ASD) or grey for $P_{corr} > 0.05$. The horizontal line within each half-violin indicates the mean balanced accuracy across all brain regions for the corresponding disorder. **B**. The mean case–control classification balanced accuracy is shown for each brain region in Desikan–Killiany brain maps for SCZ (**i**) and BP (**ii**). Only statistically significant brain regions ($P_{corr} < 0.05$, corrected across 82 regions) are shaded, and the total number of significant brain regions are indicated for SCZ (34) and BP (4). **C**. We evaluate how a given dynamical property measured across the entire brain can distinguish cases from controls per disorder $A_{feature}$ (cf. Fig 1D[ii]). For the 11 time-series features which yielded brain maps that significantly distinguished cases from controls in SCZ and/or ASD ($P_{corr} < 0.05$, corrected across 25 time-series features), the balanced accuracy is shown across all 100 test folds per feature (such that each point indicates one fold). **D**. The feature similarity score, $|\rho_{feature}|$, between each pair of the 11 features from **C** is visualized as a heatmap, revealing four distinct clusters of features based on their values measured across all brain regions from all control participants, along with the UCLA CNP SCZ group and ABIDE ASD group. **E**. We examine how the combination of all 25 time-series features measured from all brain regions can distinguish cases from controls across disorders in $A_{uni\_combo}$ (cf. Fig 1D[iii]). To compare the performance of this combination classifier with that of individual regions and time-series features, the mean balanced accuracy is shown for all brain regions (upper row, labeled $A_{region}$), the combination classifier (middle row, labeled $A_{uni\_combo}$), and time-series features (bottom row, labeled $A_{feature}$). Lines are included as a visual guide to aid comparison. Colored dots indicate a classification model with significant balanced accuracy ($P_{corr} < 0.05$); gray dots indicate models that are not significant ($P_{corr} > 0.05$).

in SCZ and BP, only one brain region (the left thalamus) was shared between the two disorders (cf. S1(B) Fig), suggesting different spatial distributions of localized alterations in each disorder. We confirmed that the case–control classification performance across brain regions in SCZ and BP was not biased by overall region volume or by gray-matter volume differences

between cases versus controls, as had been previously posited [11] (cf. S2 Fig and S2 Table). We also confirmed that the intra-regional dynamics of all significant brain regions distinguished SCZ and BP cases from controls better than a classifier trained solely on age and sex for each disorder (results shown in S3 Table), indicating the intra-region dynamical features are capturing disorder-relevant differences beyond these basic demographic features (cf. S3(E) Fig; S3 Table). However, we found that including age, sex, and head motion (quantified as mean framewise displacement, or FD) improved classification performance for some brain regions in SCZ and BP—particularly for SCZ, for which mean FD was significantly higher than controls (cf. S3(C) and S3(D) Fig). While this does not necessarily indicate that intra-regional dynamics are biased by these confounding variables, one should interpret the biological ramifications of region-specific findings with caution given the ability of confound variables to distinguish diagnostic groups.

## Analysis $A_{feature}$: Assessing changes in brain-wide maps of individual dynamical properties across disorders

In this section, we asked whether an individual time-series feature of intra-regional dynamics measured across all brain regions could statistically distinguish cases from controls. Strong classification performance in this representation would suggest that the given property is altered in one or more brain regions across cases of the corresponding disorder. To address this, we fit a separate classification model for each time-series feature, computed across all brain regions, as schematically depicted in Fig 1D(ii). As shown in Fig 2C, between SCZ and ASD, brain-wide maps of 11 out of the 25 intra-regional time-series features yielded significant balanced accuracies in at least one of the four disorders (one-tailed permutation test $P_{corr} <$ 0.05, corrected across 25 features; see S4 Table for all balanced accuracy results). Notably, the average feature classification performance was generally lower in ASD than in SCZ, and after adjusting for multiple comparisons, no time-series features significantly distinguished BP or ADHD cases from controls. Three features distinguished both SCZ and ASD cases from controls with significant balanced accuracies: the BOLD SD, the frequency corresponding to the centroid of the power spectral density ('centroid_freq'), and the first $1/e$ crossing of the linear autocorrelation function ('ACF_timescale'). By contrast, seven additional time-series features individually distinguished SCZ cases, and one additional feature distinguished ASD cases from controls; the extent of feature overlap is visually summarized in S4(A) Fig.

In order to summarize the types of time-series properties that were informative of SCZ and/or ASD diagnosis, we computed a similarity index $|\rho_{feature}|$ (defined as the absolute Spearman rank correlation between feature values measured from all brain regions in all control participants together with the UCLA CNP SCZ and ABIDE ASD groups) between all pairs of the 11 total significant features in Fig 2C (see S5 Table for a list of $|\rho_{feature}|$ values for all pairs of features). As shown in Fig 2D, we found one cluster of features that are generally sensitive to the linear self-correlation structure of a time series. This feature cluster notably included fALFF, which exhibited similar behavior across the dataset as the first $1/e$ crossing of the linear autocorrelation function ('ACF_timescale', $|\rho_{feature}| = 0.65$), the power in the lowest 20% of frequencies ('low_freq_power', $|\rho_{feature}| = 0.63$), and the centroid of the power spectral density ('centroid_freq', $|\rho_{feature}| = 0.58$). Some features in this cluster are sensitive to nonlinear structure in the time series, such as the entropy computed over probabilities of patterns over two time-steps ('entropy_pairs'), which significantly distinguished SCZ cases from controls. The 'periodicity' feature is also derived from the linear autocorrelation function, capturing the first local maximum of the ACF that meets criteria described in [79], displaying similar behavior to the features in the linear autocorrelation cluster (all $0.4 \leq |\rho_{feature}| \leq 0.55$). By contrast, the

'DFA' feature (which implements detrended fluctuation analysis to estimate the timescale at which the sharpest change in the scaling regime occurs [110]) was largely unrelated to all ten other features (all $|\rho_{\text{feature}}| < 0.2$), and was the second highest performing feature in SCZ alone ($64.1 \pm 13.3\%$, $P_{\text{corr}} = 8 \times 10^{-3}$). In general, the linear features shown in Fig 2D collectively exhibited better classification performance in SCZ and ASD than most nonlinear properties we examined, such as the time reversibility statistic 'trev' that measures nonlinear autocorrelation structure (all mean balanced accuracy values < 57% across disorders).

The SD and mean stood out as other high-performing features, which is notable given their sensitivity to the raw rs-fMRI BOLD time series distributions—in contrast to the other 23 features we evaluated, which were all computed from $z$-score normalized time series to capture properties of the underlying dynamics. Consistent with prior work [106], the BOLD SD was the top-performing feature for both SCZ ($69.8 \pm 11.7\%$, $P_{\text{corr}} = 1 \times 10^{-4}$) and ASD ($59.5 \pm 4.4\%$, $P_{\text{corr}} = 1 \times 10^{-6}$); by contrast, the mean rs-fMRI BOLD signal only distinguished SCZ cases from controls ($63.5 \pm 12.7\%$, $P_{\text{corr}} = 0.01$). BOLD SD has previously been reported as a powerful biomarker across diverse applications [111], including in case–control classification for SCZ [112, 113] and ASD [114]. However, brain-wide BOLD SD has also been quantitatively linked to head motion in the scanner [115, 116], and both SCZ and ASD groups showed greater head movement than controls (cf. S5(A) Fig and S6 Table). We therefore computed the Pearson correlation coefficient between mean framewise displacement (FD) and BOLD SD to assess their linear relationship, finding significant associations at the whole-brain level for ASD but not for SCZ (cf. S5(B) Fig and S7 Table)—suggesting different underlying biological mechanisms driving neural signal variability in the two disorders.

In summary, the $A_{\text{feature}}$ analysis revealed that brain-wide maps of the BOLD SD and properties sensitive to linear autocorrelation structure could distinguish cases from controls in SCZ and ASD, performing well compared to more complex and/or nonlinear statistics. This supports the general use of linear time-series features—like the centroid frequency of the power spectral density ('centroid_freq') or timescale of the autocorrelation function ('ACF_timescale')—to parsimoniously summarize relevant properties of rs-fMRI dynamics, while also highlighting the potential utility of the SD to clarify disorder-relevant differences in BOLD signal variance measured across brain regions.

## Analysis $A_{\text{uni\_combo}}$: Combining multiple brain regions and temporal properties to represent whole-brain fMRI dynamics across disorders

In contrast to the previous two reduced representations of intra-regional dynamics across the brain—in either a single region ($A_{\text{region}}$) or of a single time-series property ($A_{\text{feature}}$)—we next asked if a more comprehensive representation that integrates multiple time-series properties of resting-state activity measured from all brain regions could improve case–control classification performance. This entails a much higher-dimensional representation of brain-wide dynamics (2050 dimensions for the UCLA CNP cohort: SCZ, BP, and ADHD, and 1200 dimensions for the ABIDE ASD cohort). Higher-dimensional input spaces are more prone to overfitting due to greater complexity from more free parameters (described as the 'curse of dimensionality' [92]), and potentially from noise accumulation in the setting of noisy features [117]. To account for potential overfitting, we applied 10-repeat 10-fold cross-validation and we report out-of-sample performance to evaluate the generalizability of a model's predictive performance [118]. An improvement in performance (assessed via balanced accuracy) in this representation would therefore suggest that multiple properties of intra-regional activity measured across multiple spatial locations are important for characterizing a given disorder, in a way that cannot be reduced to a single region or single time-series feature. To investigate this

question, we concatenated all intra-regional features computed from the whole brain into a single 'combination' classifier model, $A_{uni\_combo}$, as depicted schematically in Fig 1D(iii).

Fig 2E shows the mean balanced accuracy using this comprehensive representation of local dynamical properties across all brain regions as singular points in the middle row ($A_{uni\_combo}$) for each disorder (see S8 Table for balanced accuracy results). We found that the $A_{uni\_combo}$ representation distinguished cases from controls particularly well for SCZ (69.5 ± 12.0%, one-tailed permutation test $P = 1 \times 10^{-7}$, Fig 2E[i]) and BP (69.8 ± 11.6%, $P = 2 \times 10^{-8}$, Fig 2E[ii]), as indicated by the points located far from chance performance (shown as dashed lines at 50%). The strong performance in SCZ is consistent with the large number of individual brain regions (34) and intra-regional time-series features (10) that yielded significant classification balanced accuracies on their own (cf. Fig 2B and 2C). By contrast, only four individual brain regions (and zero individual time-series features) significantly distinguished BP cases from controls, suggesting that BP is better characterized by the combination of multiple types of alterations to BOLD dynamics in multiple brain areas. This combination classifier did not statistically differentiate ADHD cases from controls (49.8 ± 8.7%, $P = 5 \times 10^{-1}$, Fig 2E[iii]), which is consistent with the lack of significant individual brain regions in $A_{region}$ or time-series features in $A_{feature}$. We tested the possibility that the null performance of the $A_{uni\_combo}$ classifier in ADHD may be at least partially attributable to noise accumulation [117], though we found no improvement in classification performance with a PCA-reduced feature space (cf. S6(B) Fig) nor with different classifier types (cf. S7 Fig and S9 Table).

The combination classifier did significantly distinguish ASD cases from controls, albeit with lower balanced accuracy than with SCZ or BP (53.6 ± 4.7%, $P = 1 \times 10^{-10}$, Fig 2E[iv]). We found similar results when we restricted our analysis to either of the two largest individual ABIDE sites (cf. S8(B) Fig and S10 Table), indicating that the lower performance cannot be entirely ascribed to the heterogeneity of the large multi-site ABIDE sample. We did find modest improvements in ASD classification performance using a nonlinear radial basis function (RBF) kernel as part of our robustness analysis (cf. S7(A) Fig, S9 Table), indicating that nonlinear feature-space boundaries better distinguished ASD cases from controls. In this high-dimensional feature space, this result is consistent with a larger dataset enabling more complex classification approaches to better capture signal, in a way that would be at higher risk of overfitting in a smaller dataset.

In summary, the $A_{uni\_combo}$ approach to quantifying fMRI dynamics allowed us to combine multiple properties of intra-regional dynamics from multiple brain regions in a way that can distinguish cases from controls, particularly for SCZ and BP, demonstrating the utility of combining whole-brain maps of multiple time-series features for a given classification problem.

## Analysis $A_{FC}$: Comparing features of inter-regional functional connectivity across disorders

While the previous three analyses focused on the BOLD dynamics of individual regions, in this section, we asked how patterns of distributed communication *between* pairs of brain regions assessed via functional connectivity (FC) capture case–control differences in each of the four disorders. We quantified pairwise communication as time-series dependencies, venturing beyond the traditional linear, contemporaneous Pearson correlation-based functional connectivity to compare its ability to distinguish cases from controls relative to thirteen other statistics of pairwise interactions (SPIs) from the *pyspi* library [25]. In total, these fourteen SPIs collectively measure different types of inter-regional coupling, including nonlinear, time-lagged, and frequency-based interactions, derived from literature including causal inference, information theory, and spectral methods (see Methods, Sec. 'Time-series feature extraction'

and Table 3 for more details). By comparing multiple different types of functional coupling, we aimed to investigate whether the Pearson correlation coefficient is comparatively a top performer, or if alternative metrics would be more sensitive to case–control differences in a given disorder. As depicted schematically in Fig 1D(iv), each SPI was computed between the rs-fMRI time series of all region–region pairs per participant, and we concatenated values across all region–region pairs as classifier inputs for each SPI (see Methods, Sec. 'Case–control classification'). Supplying the full region–region pair space to the linear SVM allows the classifier to learn global properties of brain connectivity through linear combinations of region pairs, like the brain-wide mean correlation [119].

We found that for SCZ and ASD, representing rs-fMRI dynamics through pairwise coupling significantly distinguished cases from controls—for five SPIs in SCZ ($P_{corr} < 0.05$, one-tailed permutation test, corrected for 14 comparisons using the method of Benjamini–Hochberg) and six SPIs in ASD. The test fold–wise balanced accuracy distribution is shown for each SPI in SCZ and ASD in Fig 3A (see S11 Table for all balanced accuracy results). By contrast, none of the fourteen SPIs significantly classified BP or ADHD cases from controls; the distribution for the top five performing SPIs in BP is visualized in S9(B) Fig, although we focus on SCZ and ASD in this section. Note that while some SPIs showed case–control differences in brain-wide average FC values (cf. S10 Fig), the global mean FC values performed similar or worse to classifiers supplied with the full region–region pair matrix (cf. S11 Fig]).

The significant performance of multiple individual SPIs suggests that inter-regional communication patterns were disrupted in both SCZ and ASD in a way that pairwise FC metrics are generally well suited to capture, consistent with prior work [120, 121]. Notably, the Pearson correlation coefficient performed well (similarly to the raw covariance matrix, cf. S12 Fig), yielding the highest classification balanced accuracy for SCZ (67.3 ± 12.6%, $P_{corr} = 1 \times 10^{-4}$, corrected across 14 SPIs) and ASD (59.5 ± 4.4%, $P_{corr} = 1 \times 10^{-9}$) (it was also among the best SPIs for BP, although this was not statistically significant; cf. S9(B[ii]) Fig). In order to better understand the types of SPIs that performed well for SCZ and ASD, we computed an empirical SPI similarity index ($|\rho_{SPI}|$) using the same method as implemented in A_feature above (i.e., computed across all controls as well as SCZ and ASD participants; cf. Results Sec. 'Analysis A_feature'). As shown in Fig 3B (and listed in S12 Table), the Pearson correlation coefficient behaved most similarly to dynamic time warping (DTW, similarity index $|\rho_{SPI}| = 0.40$), which incorporates shifts and dilations between a pair of time series [34, 122] and has previously been suggested as a promising alternative to standard correlation for rs-fMRI analysis [35]. Other top-performing SPIs included directed information ('DI', [123]) and $\Phi^*$ [124], which captured different types of pairwise coupling compared to the Pearson correlation coefficient and DTW, as shown in Fig 3B, and warrant further investigation in future work. We found similar results when assessing multi-diagnosis classification performance in the UCLA CNP cohort (cf. S13 Fig). Ultimately, the Pearson correlation coefficient was both the top-performing SPI based on case–control classification and the most parsimonious, underscoring the utility of measuring linear, synchronous inter-regional coupling under a Gaussian assumption to capture disorder-relevant pairwise dependence structures from rs-fMRI.

## Analysis A_FC_combo: Integrating local regional dynamics with pairwise coupling into one classification model across disorders

Having demonstrated that multiple statistical representations of fMRI time series—based on intra-regional dynamics and inter-regional coupling—can distinguish cases from controls in

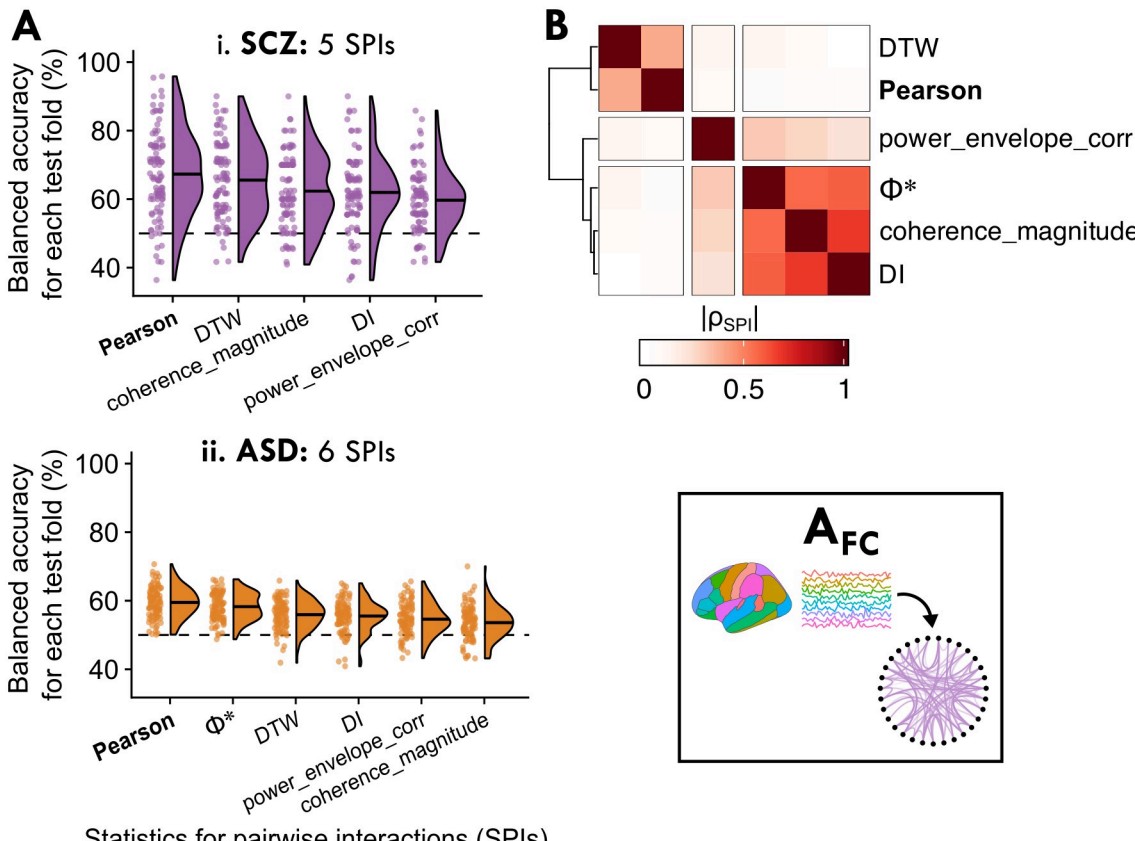

**Fig 3. Representing brain activity as the set of all pairwise functional connectivity strengths, $A_{FC}$, can significantly distinguish cases from controls, with the classical Pearson correlation coefficient (capturing linear contemporaneous coupling) a top performing metric. A.** For the 6 SPIs features which yielded brain maps that significantly distinguished cases from controls in SCZ and/or ASD ($P_{corr} < 0.05$, corrected across 14 SPIs), the balanced accuracy is shown across all 100 test folds per SPI (such that each point indicates one fold). The Pearson correlation coefficient is annotated in boldface for easier reference. **B.** The SPI similarity score, $|\rho_{SPI}|$, is visualized between each pair of the 6 SPIs from **A** as a heatmap, revealing three clusters of SPIs with similar behavior on the dataset (based on their outputs across all region–pairs and in all control participants, as well as the UCLA CNP SCZ group and ABIDE ASD group). As in **A**, the Pearson correlation coefficient annotation is shown boldface.

each disorder, we next asked whether a unified representation combining intra-regional and inter-regional properties would offer complementary information about rs-fMRI activity that could better distinguish cases from controls. To test this, for each SPI, we concatenated the FC matrix ($A_{FC}$) with the full region × univariate feature matrix ($A_{uni\_combo}$), forming the basis for analysis $A_{FC\_combo}$ as depicted schematically in Fig 1D(v). We then evaluate how well each SPI *plus* the full $A_{uni\_combo}$ matrix of local dynamics could distinguish cases from controls in each disorder.

As shown in Fig 4A, combining local dynamics with pairwise coupling allowed us to classify cases from controls with significant balanced accuracy for all 14 SPIs in SCZ, BP, and ASD ($P_{corr} < 0.05$, correcting for 14 SPIs; with the exception of phase lag index, 'PLI', for BP; see S13 Table for full results). None of the $A_{FC\_combo}$ models yielded significant balanced accuracies for ADHD after correcting for multiple SPIs (with $P_{corr} < 0.05$). The Pearson correlation coefficient remained a top-performing SPI (as in $A_{FC}$, cf. Fig 3B), yielding classification performance of $71.0 \pm 12.4\%$ in SCZ ($P_{corr} = 2 \times 10^{-8}$) and $59.4 \pm 4.5\%$ in ASD ($P_{corr} = 2 \times 10^{-8}$), as shown in Fig 4A. The $A_{FC\_combo}$ classifier using the Pearson correlation coefficient was also a

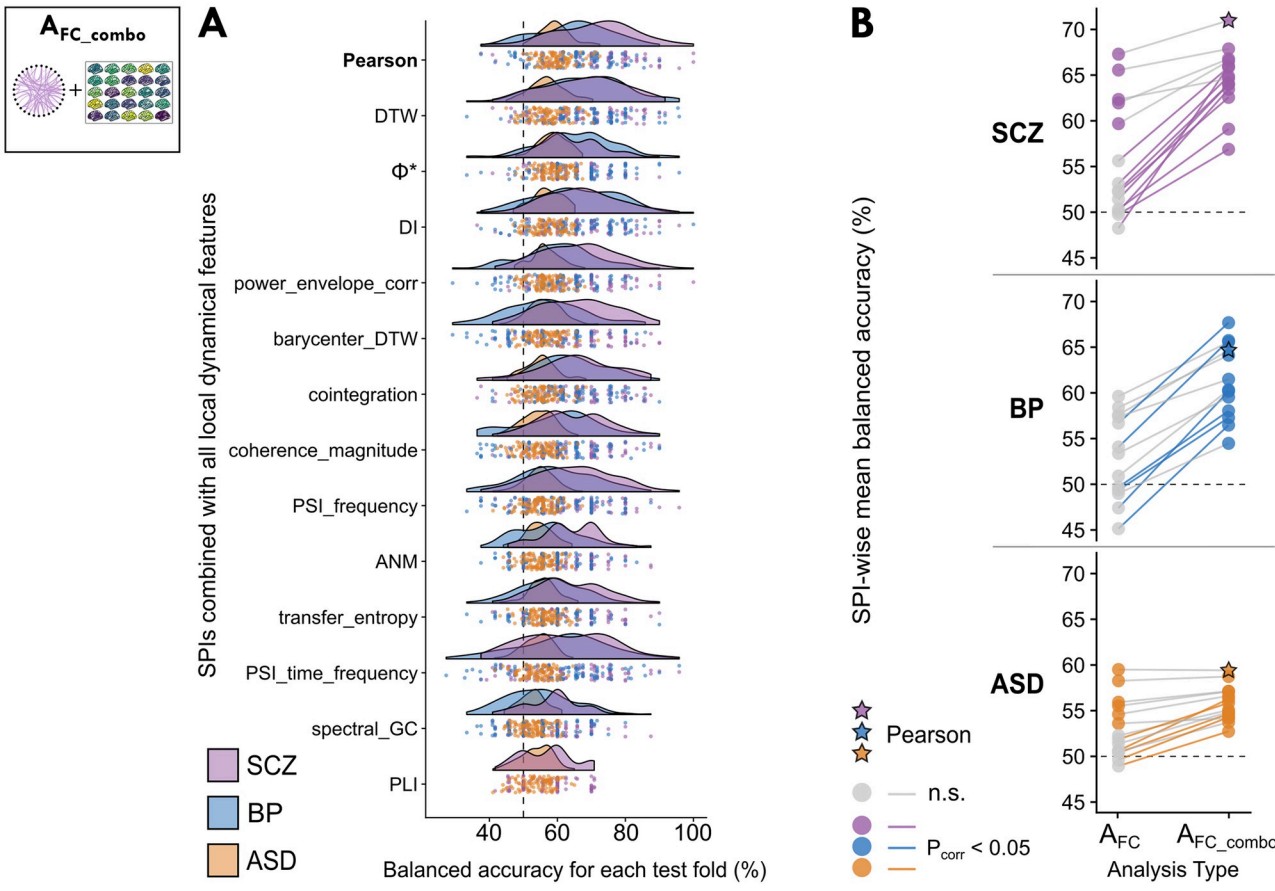

**Fig 4. Adding statistics of local regional dynamics enhances case–control classification performance for pairwise FC metrics.** Here we evaluate the effect of including brain-wide maps of local regional dynamics with pairwise coupling data for each SPI on case–control classification performance. By concatenating the full univariate region × feature matrix from $A_{uni\_combo}$ with the full set of brain region pairs per SPI from $A_{FC}$, this yielded a set of 14 linear SVM classifiers termed $A_{FC\_combo}$. **A.** The balanced accuracy is shown in raincloud plots for all 100 test folds per SPI (such that each point indicates one test fold). The Pearson correlation coefficient is annotated in boldface for easier reference. **B.** The mean balanced accuracy is shown for each SPI on its own (left, $A_{FC}$) and with the inclusion of univariate region × feature data (right, $A_{FC\_combo}$). Points are shaded in with color to indicate whether the corresponding balanced accuracy was significant ($P_{corr} < 0.05$, corrected across 14 SPIs) or not significant (gray) relative to the corresponding null distribution (cf. Methods, Sec. 'Case–control classification'). Each line corresponds to one SPI to visually guide comparison across representation types. Lines are shaded darker to indicate a significant difference in SPI performance with the inclusion of localized dynamics (corrected two-sided $t$-test $P_{corr} < 0.05$, corrected across 14 SPIs; cf. Methods, Sec. 'Case–control classification'). Stars indicate the balanced accuracy for the Pearson correlation coefficient for easier reference.

top performer in BP ($64.7 \pm 10.9\%$, $P_{corr} = 8 \times 10^{-5}$), although DTW, $\Phi^*$, and DI exhibited marginal improvements over the Pearson correlation (cf. Fig 4A). The relatively tight distributions over test folds (shown as violin plots in Fig 4A) highlight the consistency of the relatively high performance of these top SPIs across out-of-sample test cases in the three disorders, particularly in SCZ and BP.

Despite involving a considerable increase in feature-space dimensionality, adding intraregional univariate dynamics from $A_{uni\_combo}$ substantially increased the number of SPI-based models (from $A_{FC}$) that could distinguish cases from controls in SCZ, BP, and ASD. This is particularly evident for BP, for which no SPIs meaningfully distinguished cases from controls on their own (cf. Fig 3A, S9B Fig)—and yet, when we also included brain-wide maps of localized activity features in $A_{FC\_combo}$, all SPIs but PLI allowed us to classify BP cases from controls with significant balanced accuracy. In order to summarize the extent to which combining

intra-regional dynamics with inter-regional coupling improved classification performance, we directly compared the mean classification performance with each SPI in $A_{FC}$ versus $A_{FC\_combo}$ in Fig 4B. This visualization revealed that all SPIs better distinguished cases from controls with the inclusion of localized dynamics, including those that yielded statistically null performance on their own in $A_{FC}$. We quantified these improvements in case–control classification performance in $A_{FC\_combo}$ using two-tailed $T$-tests, corrected for resampled cross-validation (see Methods Sec. 'Case–control classification', [98]; all $T$-test results are presented in S14 Table). As shown by the colored lines in Fig 4B, the improvement with $A_{FC\_combo}$ was significant (after correcting for 14 SPI comparisons per disorder) for 9 SPIs in SCZ, 6 SPIs in BP, and 5 SPIs in ASD.

The addition of intra-regional time-series features did not just improve the performance of significant $A_{FC}$ models, but also elevated the performance of many non-significant SPIs to classify cases from controls well across disorders. For example, additive noise modeling ('anm')—a measure of nonlinear dependence that tests causality from one brain region to another with a Gaussian process [125]—did not distinguish cases from controls well on its own (all balanced accuracies < 52% averaged across test folds), but the inclusion of local uni-variate dynamics boosted its performance to 64.0 ± 8.6% in SCZ, 57.3 ± 9.4% in BP, and 54.7 ± 4.3% in ASD (all balanced accuracy $P_{corr} < 3.0 \times 10^{-3}$ and $T_{corr} > 2.4$). Similar improvements were observed across disorders for the barycenter ('barycenter'), a measurement of the center of mass of the rs-fMRI BOLD time series between the pair of brain regions, with the barycenter magnitude reflecting the extent of dynamic coupling [126, 127]. While most of these significantly improved SPIs yielded mid-ranking balanced accuracies in $A_{FC\_combo}$, in BP, the two top-performing SPIs—DTW (67.7 ± 12.0%, $P_{corr} = 2 \times 10^{-6}$) and $\Phi^*$ (65.7 ± 11.0%, $P_{corr} = 2 \times 10^{-8}$)—exhibited among the largest improvements with the inclusion of intra-regional univariate dynamics (both $T_{corr} = 2.7$, $P_{corr} = 0.03$). However, SPIs that performed well for SCZ and ASD on their own ($A_{FC}$), such as the Pearson correlation coefficient, generally did not show large margins of improvements with the inclusion of local univariate dynamics (all $T_{corr} < 1.7$ and $P_{corr} > 0.09$)—suggesting that such high-performing SPIs capture relevant case–control information that overlaps more with that of local univariate dynamics than low-performing SPIs.

In summary, we found that including intra-regional time-series features generally improved the case–control classification performance of SPIs, particularly for those which did not separate cases from controls well on their own in $A_{FC}$. This underscores the benefits of a unified representation that combines local intra-regional dynamics and pairwise inter-regional coupling to simultaneously capture complementary aspects of brain activity changes across clinical settings.

## Identifying which representation type(s) are optimally suited to capture case–control differences in each disorder

In previous sections, we evaluated a different number of models within each representation type—e.g., 25 intra-regional features in $A_{feature}$ versus 1 model for $A_{uni\_combo}$ versus 14 SPIs for $A_{FC}$. These differences in model numbers (and model complexity) from $A_{region}$ through to $A_{FC\_combo}$ make it challenging to fairly compare the relative strengths of each representation in summarizing case–control differences in rs-fMRI dynamics. To address this, we implemented a strategy to build a single model incorporating the statistical information of a given representation type, corresponding to the five ways of quantifying dynamical properties from an rs-fMRI dataset evaluated above. For each cross-validation fold, we identified the top-performing model based on in-sample training performance ($A_{region}$ or $A_{feature}$) or performance

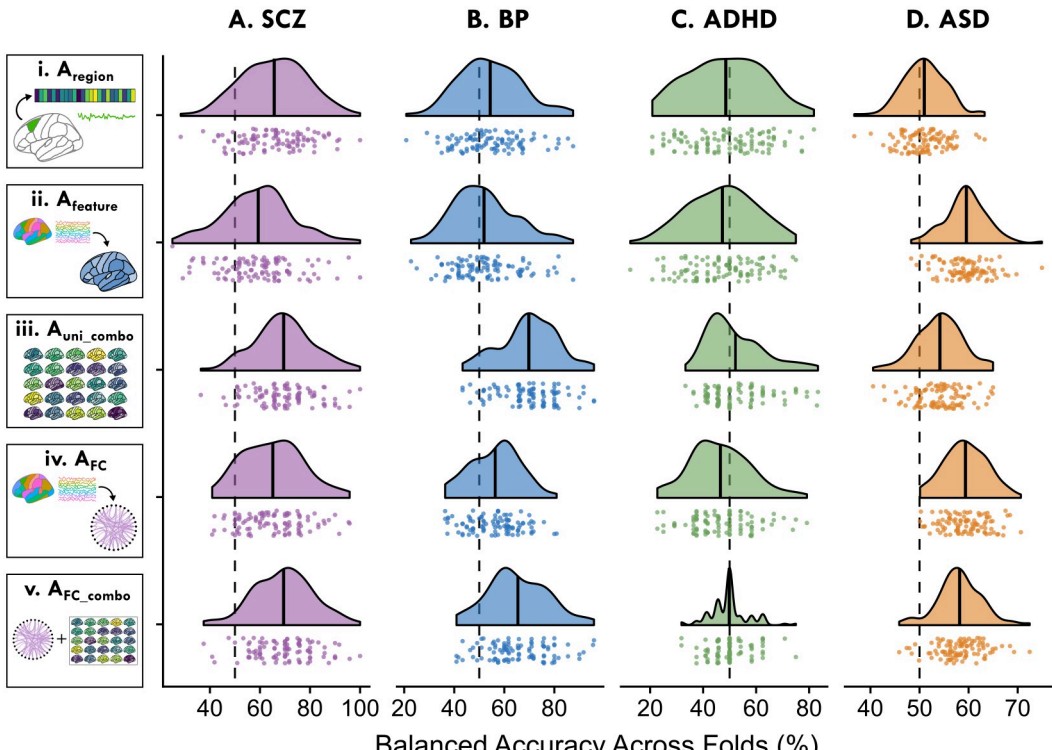

**Fig 5. Direct comparison of model performance across representations and neuropsychiatric disorders.** For SCZ (**A**), BP (**B**), ADHD (**C**), and ASD (**D**), we show the distribution of the best-performing model for each of 100 test folds across the five representations. In $A_{region}$ (**i**) and $A_{feature}$ (**ii**), we identified the model that yielded the top in-sample training balanced accuracy per fold and retained its out-of-sample test balanced accuracy. In $A_{uni\_combo}$ (**iii**), as there was only one model for comparison, the distribution shows the 100 test folds for this model. For $A_{FC}$ (**iv**) and $A_{FC\_combo}$ (**v**), we first performed a PCA and retained the first 10 PCs per SPI, then identified which PC-space SPI yielded the top in-sample training balanced accuracy per fold. Upon identifying that SPI, the full feature-space SPI was evaluated with the linear SVM classifier to measure the out-of-sample balanced accuracy for the corresponding test fold. For each half-violin plot, the horizontal line indicates the mean balanced accuracy across all 100 test folds for the given representation type (row) and disorder (column).

in a reduced feature space of the first 10 PCs (see Methods). This strategy enabled us to directly compare classification performance on unseen data from just one model per representation across test folds.

In Fig 5, we show the distribution of classification balanced accuracy values across the 100 test folds per disorder as raincloud plots. As shown in Fig 5(iii), $A_{uni\_combo}$ exhibited the best overall classification performance for SCZ (Fig 5A) and BP (Fig 5B); all results for this analysis are in S15 Table. This is particularly evident for BP, in which selecting one brain region or one intra-regional time-series feature yielded an average classification balanced accuracy of 54.4% or 52.2%, respectively—yet the combination of local dynamics across brain regions in $A_{uni\_combo}$ yielded an average 69.8% balanced accuracy. Selecting one brain region per test fold yielded 65.8% balanced accuracy in SCZ, which is surprisingly high for a disorder generally characterized by distributed dysfunction [11–13], and this was surpassed by the $A_{uni\_combo}$ average of 69.5% balanced accuracy. By contrast, in ASD, selecting one representative intra-regional time-series feature per test fold outperformed all other representations, yielding 59.5% balanced accuracy on average—an effect driven primarily by the BOLD SD, which was selected in 99 out of 100 cross-validation folds, consistent with results shown for ASD in

$A_{region}$ (Fig 2C and 2D). Since $A_{uni\_combo}$ included all of the BOLD SD data from $A_{feature}$, the drop in classification performance is surprising, underscoring how uniquely informative the SD was for ASD—with the caveat that this effect may be driven in part by head motion (cf. S5 Fig).

Comparing these univariate models with $A_{FC}$ (Fig 5iv) and $A_{FC\_combo}$ (Fig 5v), we found that picking the best representative SPI in $A_{FC\_combo}$ (69.4% average balanced accuracy) yielded comparable balanced accuracy to that of $A_{uni\_combo}$ (69.5%) in SCZ. This suggests that for any given SPI in $A_{FC\_combo}$, the local univariate dynamics are a key driver in the improved classification performance for SCZ. A similar relationship was observed in BP (Fig 5B), in which $A_{uni\_combo}$ (69.8% average balanced accuracy) actually surpassed the classification performance of the representative subset of SPIs in $A_{FC\_combo}$ (65.4% average). Moreover, for ASD (Fig 5D), the representative intra-regional feature subset—predominantly comprised of the BOLD SD—performed similarly to that of $A_{FC}$ (59.4% average balanced accuracy) and $A_{FC\_combo}$ (58.2%). This comparison highlights the utility of localized, intra-regional dynamics in distinguishing cases from controls across disorders, with room for future work optimizing feature selection and classifier parameters in the case of models with larger feature spaces (e.g., $A_{FC}$ and $A_{FC\_combo}$).

## Discussion

There are myriad ways in which a data analyst can extract interpretable feature-based representations of dynamical structures contained in a multivariate time series, like those measured with rs-fMRI. From this statistical smorgasbord, a given researcher typically chooses a set of dynamical properties to study for a given problem subjectively, such as a set of pairwise Pearson correlation coupling strengths or fALFF in a given set of brain regions. The lack of systematic methodological comparison thus leaves it unclear from any given study whether these chosen dynamical properties are optimal, or whether alternative—and potentially simpler and more interpretable—statistics may outperform those reported. To address these concerns, here we introduced a systematic comparison of feature-based representations of rs-fMRI time series, based on localized intra-regional dynamics, pairwise inter-regional coupling, and their combination, allowing us to systematically capture and compare different facets of rs-fMRI dynamics. Our results demonstrate the benefits of such comparison by identifying the most parsimonious and informative types of structures that are relevant for a given application, revealing disorder-specific signatures across neuropsychiatric groups. Our work provides a methodological foundation for systematically invoking representative features from a rich interdisciplinary literature on time-series analysis [23, 25] to determine the most appropriate way(s) of summarizing interpretable dynamical structures in MTS datasets. This approach is generalizable to a wide range of problems across neuroimaging modalities and applications, as well as to a vast array of science and industry problems in which complex time-varying systems are measured and analyzed.

The five statistical representations of MTS evaluated here can be investigated individually to gain insights into the types of local dynamics and pairwise dependencies that are optimally suited for a given application. For example, by comparing model performance within $A_{region}$ (Results Sec. '$A_{region}$'), we found that the dynamics of many individual brain regions could distinguish cases from controls in SCZ, and to a lesser extent in BP. This is a striking finding, given that such complex neuropsychiatric disorders are believed to arise from distributed dysfunction across brain networks rather than localized changes to individual regions [12, 14, 109, 128]—which suggests that our expanded breadth of rs-fMRI time-series features combined into a unified statistical representation for each brain region is informative of disorder-

specific alterations. Examining dynamics within individual brain regions enables spatial interpretability through visualizing brain-wide maps of classification performance, providing a clear region-by-region picture of activity disruptions. The ability to characterize changes in BOLD dynamics at the level of individual regions has been key to addressing questions about regional differences in response to spatially targeted brain stimulation [22, 129]. It also provides a clearer way to test molecular hypotheses about regional disruption in disorders, which can be compared with rich multimodal region-level atlases spanning morphometry, cortical hierarchy, and multi-omics [30, 130–132] to more deeply characterize the physiological underpinnings of disease-relevant changes in future work. This systematic approach recapitulated prior results of individually informative brain regions—like medial occipital regions in SCZ [133–135] and subcortical regions in BP [136, 137]—while also identifying novel region-specific changes in local dynamics that generate new hypotheses for future work. While we did not find significant region-specific patterns of BOLD alterations in ADHD or ASD, this does not preclude the possibility that different parcellation schemes and/or different intra-regional features (beyond the 25 examined here) could better elucidate disorder-specific alterations [138].

Looking across the features we compared in this study, our findings generally support the use of linear time-series analysis techniques that are commonly used for rs-fMRI data analysis, both at the level of individual brain regions and at the level of pairwise coupling—while simultaneously identifying novel high-performing metrics that warrant future investigation, including directed information to capture asymmetric information flows. In $A_{feature}$ (Results Sec. '$A_{feature}$'), we found that whole-brain maps of individual intra-regional dynamical properties related to linear autocorrelation (including Fourier power spectrum structure, like the centroid of the power spectrum) were top performers in distinguishing SCZ and ASD cases from controls (see Fig 2C). For example, the 'ACF_timescale' statistic distinguished ASD cases from controls while the fALFF did not (despite their high empirical correlation, as has been noted in previous work [33]), demonstrating that algorithms derived from theoretically similar foundations can perform differently in a given application. Interestingly, the strong performance of this timescale feature might be related to previous work showing changes to intrinsic neural timescales in SCZ [139, 140] and ASD [140, 141]. The nonlinear intra-regional features we examined (such as the time reversibility statistic 'trev') did not enable us to distinguish cases from controls in any disorder, consistent with the view that rs-fMRI BOLD dynamics (which are noisy and sparsely sampled in time) are well approximated by a linear stochastic process such that methods aiming to capture more complex (e.g., nonlinear) dynamical structures may not be beneficial at this timescale, as has recently been proposed [142, 143].

The BOLD SD stood out relative to the other top-performing intra-regional features, as it is sensitive to the raw time-series values—while all other features (besides the mean) were computed after $z$-score normalization, in order to focus on underlying dynamics of the time series in a way that does not depend on the measurement scale. We found that whole-brain maps of the BOLD SD were informative of SCZ and ASD diagnosis relative to controls (see Fig 2C), expanding upon our previous work showing that the BOLD SD outperformed other univariate time-series features for SCZ [106]. Changes to BOLD signal variability have previously been described in both SCZ [112, 113] and ASD [114], with prior work underscoring its utility and reliability as an rs-fMRI statistic [144]. More broadly, regional BOLD SD alterations have been reported in settings ranging from healthy aging [145, 146] to Alzheimer's disease [147], and it is hypothesized that BOLD signal variance is linked to functional integration [148] as well as numerous molecular and cytoarchitectural properties [111]. While we demonstrated that BOLD SD was generally unrelated to head movement in SCZ we are cautious with our interpretations for ASD given the strong positive associations between SD and head movement

(a non-neural confound) in the ABIDE cohort; this warrants further investigation and clarification in a future study.

When we combined multiple properties of intra-regional dynamics across the whole brain in $A_{uni\_combo}$, we found improved case–control classification performance relative to either representation on its own ($A_{region}$ or $A_{feature}$) in SCZ and BP—consistent with disruptions to brain dynamics that are both spatially distributed and multifaceted [15, 58, 149]. By contrast to the improved performance with the expanded $A_{uni\_combo}$ representation for SCZ and BP, reduced representations (i.e., individual brain regions or individual time-series features) better distinguished ASD cases from controls (see Figs 2E and 5D). In ASD, the BOLD SD was the dominant high-performing feature among the 25 we compared, emerging as the top-performing metric in 99 out of 100 training validations (see Fig 5D). Although the BOLD SD was contained within the combined region × feature matrix evaluated in the combination classifier $A_{uni\_combo}$, increasing the number of input attributes to a classifier does not necessarily improve its performance due to overfitting beyond the 'latent dimensionality' of the dataset, defined as the number of meaningful variables inferred from the data that capture underlying essential patterns [150, 151]. This suggests that aside from the BOLD SD, the other local univariate time-series features contributed less disease-relevant information to the linear classifier, thereby demonstrating that systematic comparison can uncover potential model simplifications to select a more parsimonious feature-based representation.

Through our comparison of SPIs in $A_{FC}$ (see Results Sec. '$A_{FC}$'), we found that pairwise coupling strengths also served as an informative way to represent dynamical structures related to case–control differences. Out of the 14 evaluated SPIs, we found that the Pearson correlation coefficient was a top-performing statistic in both SCZ and ASD, suggesting that linear time-series analysis methods are overall well-suited for capturing the salient dynamical properties of rs-fMRI MTS [30, 33, 152]. Our comparisons also highlighted interesting alternative statistics with different behavior but similarly high performance to the Pearson correlation coefficient, such as DI [123, 153] and $\Phi^*$ [124, 154]. These SPIs involve more conceptually and computationally complex formulations of pairwise interactions that are seldom applied to fMRI datasets, although we have previously found that alternative metrics like DI out-performed the Pearson correlation coefficient in an fMRI-based problem [25]. While there was little evidence that increasing the complexity in an FC measurement beyond the Pearson correlation coefficient substantially improved SCZ or ASD classification here, we did observe modest improvements in BP classification using each of DTW, $\Phi^*$, and DI (with the inclusion of local univariate dynamics) instead. The alternative FC metrics examined here could be explored in future work using data with higher temporal resolution and signal-to-noise ratios (e.g., from MEG [32, 155])—a setting in which more complex types of interactions may be measurable (e.g., that are nonlinear and/or time-lagged).

Consistent with complex and spatially distributed disruptions to rs-fMRI activity, we found that all 14 SPIs better distinguished cases from controls for SCZ, BP, and ASD when we also included brain-wide maps of all 25 intra-regional time-series features in $A_{FC\_combo}$ (see Results Sec. '$A_{FC\_combo}$'). This finding is in line with previous work demonstrating that combining local and pairwise properties of rs-fMRI data can synergistically improve classification performance [11, 38, 39, 156], supporting the notion that intra-regional activity and inter-regional coupling provide complementary information about disorder-relevant brain dynamics. While we cover an extensive space of dynamical statistics here, future applications might combine local dynamics with inter-regional coupling using network properties, local–global coupling, geometric embedding techniques like regional homogeneity (ReHo), or higher-order interactions. For example, [157] and [158] demonstrated the benefits of quantifying 'glocal' measures

[159] of synchrony between individual regions and whole-brain networks for distinguishing wakefulness from anesthetic states.

Future applications of this systematic framework might also consider more nuanced data fusion techniques beyond simple matrix concatenation to properly combine heterogeneous input data types (e.g., multiple brain regions; local dynamics versus pairwise coupling) into one classifier [160, 161]. Our modular approach incorporates incremental complexity from individual brain regions up to integrated whole-brain maps of local dynamics and pairwise coupling, such that results are interpreted based on the overall performance of a given model. While we did not explore feature importance scores due to intrinsic issues with multicollinearity among input features, future work could incorporate dimensionality reduction techniques to mitigate this collinearity and specifically interpret individual feature weights (e.g., linear SVM coefficients) to compare relative contributions of different brain regions and/or temporal signatures to a given decision boundary. More broadly, techniques to distill down the feature space will be of great utility to analyzing complex system MTS in general, both in terms of computational demand and mitigating noise accumulation.

While it is not straightforward to compare classification performance across disorders given differences in sample size and acquisition sites [117], we observed that SCZ cases were distinguished from controls with higher balanced accuracy across representations than the other three disorders. By contrast, for ADHD, none of the five representations we evaluated yielded significant case–control balanced accuracies. Inverse probability weighting boosted classifier performance beyond predicting the majority class in many models in ADHD (cf. S14 (A) Fig), although our robustness analysis across classifier types and hyperparameter tuning suggests that poor classification performance is attributable to high heterogeneity in a small sample size. Indeed, the ADHD sample ($N = 39$) was smaller than that of SCZ ($N = 48$), BP ($N = 49$), or ASD ($N = 513$), though previous studies have reported conflicting findings about the effect of sample size on classification performance [17, 162, 163]. It is possible that the ADHD group is comprised of individuals with less severe symptoms than that of SCZ or BP— which could decrease classification performance with rs-fMRI features [164]—although we did not explicitly incorporate ADHD assessment or medication data in the scope of this analysis. In support of this possibility, there was no difference in head motion between the ADHD cases and controls, which is surprising for a disorder generally characterized by hyperkinesis [165] and reports of greater head motion in the scanner [166, 167]. Multiple publications have reported that head motion in the scanner is linked to symptom severity in ADHD [167–169], and [170] found in a meta-analysis that few neuroimaging studies detected significant ADHD–control alterations in the absence of group head motion differences.

We found several intra- and inter-regional properties of rs-fMRI dynamics that significantly distinguished ASD cases from controls, although the corresponding balanced accuracy values were generally lower than that of SCZ and BP. There are a few potential explanations for why ASD cases were less distinguishable from healthy controls than SCZ or BP in general. The ABIDE dataset is approximately ten times larger than any of the disorder–control combined groups in the UCLA CNP cohort [47, 48], and prior studies have demonstrated an inverse relationship between sample size and classification performance using ABIDE data [171–175]. Notably, we included ASD and control participants from all ABIDE sites and did not explicitly account for imaging site in our classification analyses. While site-specific effects appear to be minimal in this dataset [69] and non-disorder factors like imaging protocol or scanner seem to contribute less to variance in resting-state activity than key brain regions [176], we did compare classifiers restricted to participants from each of the two largest ABIDE sites, finding minimal differences to those trained on the full ABIDE dataset. However, future work could evaluate the effect of multi-site integration techniques such as ComBat

harmonization [177, 178] on intra- and inter-regional feature performance. Another difference with the ASD group relative to the other disorders (from the UCLA CNP cohort) is that homotopic region pairs were consolidated into one bilateral region as part of the competition data provided by [60], which could obscure relevant hemisphere-specific changes to localized dynamics in ASD [179, 180]. It is possible that these populations exhibited changes to neural functional architecture that were more heterogeneous and individual-specific [162, 181, 182], making it difficult to identify an effective SVM decision boundary across representations— underscoring the value of ongoing work with normative modeling [183].

The goal of this study was to compare within and across interpretable representations of rs-fMRI dynamics, with a focus on results obtained with the linear SVM classifier for simplicity. We explored alternative classifiers that implement SVM with an RBF kernel or L1 ('LASSO') regularization, random forest, or gradient boosting (cf. S7 Fig) and did not observe general performance improvements, although it is possible that combining other nonlinear classifiers with hyperparameter optimization could more sensitively discriminate among features in a given disorder if nonlinear boundaries are present. Maximizing case–control classification performance was not an explicit aim of this study, although we note that our findings sit within the range of recent rigorous large-scale classification studies [184–186]. Future work could expand upon this generalizable framework to optimize performance in a given application setting.

In the '$p \gg n$' [187] large feature space setting with $A_{uni\_combo}$, $A_{FC}$, and $A_{FC\_combo}$, alternative regularization may be beneficial—which could be accomplished through more exhaustive hyperparameter tuning and/or different regularization approaches, such as graphical lasso [188]. Future work could also systematically evaluate ways to reduce (spatial) dimensionality in a biologically informative manner (e.g., condensing region pairs into canonical functional networks [189] or applying similarity network fusion [161, 190]). While this work presents a systematic methodological framework that is flexible and generalizable across domains, future studies aiming to identify disorder-specific biomarkers will need to evaluate their findings in external validation datasets to thoroughly assess the validity of a given brain region or time-series feature. Moreover, future work could extend beyond the binary case–control classification paradigm to examine transdiagnostic versus disorder-specific properties of resting-state dynamical structure [191, 192].

Here we focused on five key ways of systematically comparing intra-regional and inter-regional features from an fMRI dataset, but many other representations of spatiotemporal data could also be investigated in future work. This includes quantifying properties of the networks defined by pairwise FC matrices (cf. highly comparative graph analysis [193]), statistics of spatiotemporal patterns (like spirals and traveling waves [194, 195]), and higher-order (beyond pairwise) dependence structures [196]. For example, statistics such as metastability (based on the standard deviation of the Kuramoto order parameter) could quantify properties of integration and segregation across brain states [197]. More broadly, future work could relax the picture of interacting spatially localized and contiguous brain regions, towards spatially distributed modes, extracted as components through dimensionality reduction [198, 199] or geometrically [200, 201]. We also focused here on representative subsets of the univariate time-series analysis literature (the *catch22* subset of over 7000 features in the *hctsa* feature library [24]) and pairwise dependence literature (14 representative SPIs from the full *pyspi* library of over 200 SPIs [25]). Note that neither the univariate features nor the SPIs measured here were specifically tailored to neuroimaging applications; for example, the *catch22* set was derived based on classification performance across 128 diverse univariate time series datasets (including beef spectrograms and yoga poses) [75]. Indeed, given the demonstrated biological heterogeneity in neuropsychiatric disorders [202], effective exploration of larger time-series

feature spaces could identify subtypes within a given disorder with distinctive dynamical profiles [203]—establishing the foundation for data-driven nosology [204]. Subsequent work could consider ways to incorporate the full sets of local and pairwise time-series properties or to derive reduced subsets with algorithmic approaches tailored to a particular dataset. Broadening the scope of comparison in these ways—of both types of dynamical structures and algorithms for quantifying them—comes with associated issues of statistical power required to reliably pin down specific effects that future work will need to carefully consider.

## Supporting information

**S1 Fig. While several brain regions exhibited altered dynamics in each case–control comparison, only a subset were significant in SCZ and BP after multiple comparisons. A**. The distribution of balanced accuracy values across test folds is shown as a raincloud plot for each significant brain region ($P_{adj} < 0.05$, corrected across 82 regions) in SCZ (**i**) and BP (**ii**). The horizontal line within each half-violin indicates the mean balanced accuracy for the corresponding brain region. **B**. The Venn diagram illustrates the number of significant brain regions for each of SCZ (purple) and BP (blue), indicating that one brain region (left thalamus) is shared between the two disorders. **C**. For each of the four disorders, regions are shaded dark to indicate $P_{adj} < 0.05$ (corrected across 82 regions for SCZ, BP, and ADHD; across 48 regions for ASD). Additionally, regions are shaded light to indicate that the nominal uncorrected $P < 0.05$. Gray shading indicates that the uncorrected $P > 0.05$ for the balanced accuracy in the given region.
(TIFF)

**S2 Fig. Region-wise classification balanced accuracy is not associated with volumetric differences across clinical groups. A**. For each brain region in the UCLA CNP dataset, the mean balanced accuracy is plotted relative to the absolute $\beta$ coefficient estimated from ordinary least squares regression of region volume on diagnosis per clinical group. Pearson correlation estimates ($R$) and corresponding $P$-values are annotated in the top right corners. **B**. As in **A**, for each brain region, the mean balanced accuracy is plotted relative to the average region volume (measured in number of voxels) across all participants in the UCLA CNP cohort. Pearson correlation estimates, $R$, and corresponding $P$-values are shown in the top right corners.
(TIFF)

**S3 Fig. BOLD fMRI feature-based representations generally improve diagnostic classification beyond age, sex, and mean FD. A**. The percentage of males (blue) and females (red) per diagnostic group is shown for the UCLA CNP cohort (upper) and the ABIDE cohort (lower). Sex proportions were compared for each disorder relative to the corresponding control group using a chi-square test, with significance level indicated as ***$P < 0.001$, **$P < 0.01$, *$P < 0.05$, n.s. $P > 0.05$. **B**. The distribution of participant ages is shown as a raincloud plot per diagnostic group for the UCLA CNP cohort (left) and the ABIDE cohort (right). The horizontal line within each half-violin indicates the mean age for the corresponding distribution. Age distributions were compared for each disorder relative to the corresponding control group using a Wilcoxon rank-sum test, with significance level indicated as ***$P < 0.001$, **$P < 0.01$, *$P < 0.05$, n.s. $P > 0.05$. **C**. The mean framewise displacement (FD) computed with the method from [71] is shown with raincloud plots for all participants in the UCLA CNP (left) and ABIDE cohorts (right). The horizontal line within each half-violin indicates the mean FD for the corresponding group. Mean FD distributions were compared for each disorder relative to the corresponding control group using a Wilcoxon rank-sum test, with significance level indicated as ***$P < 0.001$, **$P < 0.01$, *$P < 0.05$. **D**. Case–control classification balanced

accuracy is shown for all 100 test folds per each disorder based on participant sex, age, or mean FD. The horizontal line within each half-violin indicates the mean balanced accuracy for the corresponding distribution. The dashed horizontal line in the background denotes 50% balanced accuracy. **E**. For each representation type (rows) and neuropsychiatric disorder (columns), the classification accuracy of each model (e.g., left pericalcarine cortex in A$_{region}$) is compared using five distinct classier inputs: (1) age + sex only, in purple; (2) age + sex with BOLD time-series features from the given model, in yellow; (3) just BOLD time-series features, in green; (4) age + sex + mean FD with BOLD time-series features, in red; and (5) age + sex + mean FD only. Each dot corresponds to one model (e.g., the left pericalcarine cortex in [i] A$_{region}$) and lines connect model types to guide visual comparison. The horizontal dashed line in each plot is included to show the chance baseline performance of 50% balanced accuracy. (TIFF)

**S4 Fig. While several brain regions exhibited altered dynamics in each case–control comparison, only a subset were significant in SCZ and BP after multiple comparisons. A**. The Venn diagram illustrates the number of significant intra-regional time-series features for each of SCZ (purple) and ASD (orange), indicating that three features are shared between the two disorders. ₈. The distribution of balanced accuracy values across test folds is shown as a rain-cloud plot for each intra-regional time-series feature that yielded a balanced accuracy with either $P_{corr} < 0.05$ (darker) or $P_{uncorr} < 0.05$ (lighter). The horizontal line within each half-violin indicates the mean balanced accuracy for the intra-regional time-series feature. (TIFF)

**S5 Fig. Head motion is generally higher in cases than in controls, but is only associated with brain-wide BOLD SD in the ABIDE cohort. A**. The mean framewise displacement (FD) computed with the method from [71] is shown with raincloud plots for all participants in the UCLA CNP (left) and ABIDE cohorts (right). The horizontal line within each half-violin indicates the mean FD for the corresponding group. Mean FD distributions were compared for each disorder relative to the corresponding control group using a Wilcoxon rank-sum test, with significance level indicated as *** $P < 0.001$, ** $P < 0.01$, * $P < 0.05$, n.s. $P > 0.05$. **B**. The brain-wide average BOLD SD is plotted against the mean FD across SCZ and control participants in the UCLA CNP cohort (upper) as well as ASD and control participants in the ABIDE cohort (lower), with the Pearson correlation estimates ($R$) and corresponding p-values shown in each plot. **B**. For each brain region, the average BOLD SD is shown across all SCZ and control participants in the UCLA CNP cohort (upper) as well as ASD and control participants in the ABIDE cohort (lower). Note that different color scales are used for the two cohorts, respectively. **D**. For each brain region, we computed the Pearson correlation between the region-wise BOLD SD and whole-brain mean FD values in the UCLA CNP cohort (upper) as well as ASD and control participants in the ABIDE cohort (lower). Pearson correlation estimates (R) are shown in brain maps, in which only brain regions for which Benjamini–Hochberg corrected $P < 0.05$ are shaded (corrected across 82 regions for UCLA CNP and 48 regions for ABIDE). Note that the same color scale is used for both cohorts. (TIFF)

**S6 Fig. Linear dimensionality reduction and regularization approaches did not improve out-of-sample classification for the univariate region × feature classifiers, A$_{uni\_combo}$. A**. For each disorder, individual scores for the first two PCs are plotted, with points colored according to diagnosis. Shaded areas reflect convex hulls encapsulating all points for each diagnostic group. Note that each PCA was computed separately for each case–control comparison, so PC1 and PC2 scores are not directly comparable across clinical groups. **B**. For each case–

control comparison, we compare the out-of-sample balanced accuracy across the 100 repeats × folds using all region × feature variables (left, dark purple) versus using only scores for the first 25 PCs (right, light purple). Points are randomly jittered along the horizontal axis in each raincloud plot to aid visualization. The horizontal line within each half-violin indicates the mean balanced accuracy for the corresponding distribution. **C**. For each case–control comparison, we compare the out-of-sample balanced accuracy across the 100 repeats×folds using default regularization (left, dark green) versus L1 ('LASSO' [90]) regularization (right, light green).
(TIFF)

**S7 Fig. Comprehensive comparison of classifier types and hyperparameter optimization supports the use of linear SVM. A**. For each of the three univariate representations—$A_{region}$, $A_{feature}$, and $A_{uni\_combo}$—the mean cross-validated balanced accuracy is shown per disorder using each of five different classifier types. Each dot corresponds to one model input type (e.g., left pericalcarine cortex in $A_{region}$) and lines connect model input types across classifiers per disorder to guide visual interpretation. The dashed horizontal lines indicate 50% balanced accuracy in all plots. **B**. For the same univariate representations as in **A**, the mean cross-validated balanced accuracy is shown for the linear SVM classifier without hyperparameter optimization (i.e., explicitly setting $C = 1$ and applying inverse probability weighting; green) or with hyperparameter optimization for the $C$ parameter and sample weighting type in purple. The dashed horizontal lines indicate 50% balanced accuracy in all plots.
(TIFF)

**S8 Fig. Classification performance is comparable within individual ABIDE consortium imaging sites.** The mean cross-validated balanced accuracy is shown with the inclusion of participants from all ABIDE sites together (blue, $N = 1091$ participants) for each of **(i)** $A_{region}$, **(ii)** $A_{feature}$, **(iii)** $A_{uni\_combo}$, **(iv)** $A_{FC}$, and **(v)** $A_{FC\_combo}$. The mean cross-validated balanced accuracy is also shown when we restricted classification analyses to each of the two largest ABIDE imaging sites: Site #20 (purple, $N = 106$ participants) and Site #5 (yellow, $N = 98$ participants). Each dot corresponds to one individual model (e.g., the Superior Frontal Gyrus in $A_{region}$) and lines connect models across ABIDE site analyses to guide visual interpretation. The dashed horizontal line marks 50% balanced accuracy in all plots.
(TIFF)

**S9 Fig. Representing brain activity as the set of all pairwise functional connectivity strengths, $A_{FC}$, can significantly distinguish cases from controls, with the classical Pearson correlation coefficient (capturing linear contemporaneous coupling) a top performing metric. A**. We compared 14 statistics of pairwise interactions (SPIs) (from *pyspi* [25], cf. Methods Sec. 'Time-series feature extraction'), as different ways of quantifying functional connectivity (FC) between pairs of brain regions. For a given SPI, each participant was represented by the set of corresponding FC values (calculated for each pair of brain regions), yielding a set of region–region pair values that can be stored as a one-dimensional vector per participant. These vectors were concatenated to yield a participant × region–pair matrix that formed the basis for case–control classification using a linear SVM. **B**. Classification results are shown as a heatmap, with rows representing SPIs that yielded significant balanced accuracy ($p_{corr} < 0.05$, corrected across 14 SPIs) in at least one disorder, and columns representing each of the four disorders. Of the 14 SPIs we evaluated, eleven significantly distinguished cases from controls in at least one disorder, and are plotted here. The Pearson correlation coefficient is annotated in boldface for easier reference. **C**. The SPI similarity score, $|\rho_{SPI}|$, is visualized between each pair of the eleven SPIs from B as a heatmap, revealing six clusters of SPIs with similar behavior

on the dataset (based on their outputs across all region–pairs and all disorders). As in B, the Pearson correlation coefficient annotation is shown boldface. **D**. The disorder similarity score, $\rho_{disorder}$, is depicted to compare the balanced accuracy values among all 14 SPIs between each pair of neuropsychiatric disorders; a large positive $\rho_{disorder}$ indicates a strong positive Spearman correlation in case–control classification performance across the 14 SPIs in the given pair of disorders.
(TIFF)

**S10 Fig. The mean FC value across all region–region pairs by SPI.** For each participant in the UCLA CNP and ABIDE cohorts, we calculated the mean FC value across all region–region pairs per SPI and show the distributions across participants as raincloud plots. The solid horizontal line in each half-violin indicates the mean balanced accuracy and the dashed horizontal axis line denotes 50% balanced accuracy. Wilcoxon rank-sum test results are indicated with ***, $P_{corr} < 0.001$; **, $P_{corr} < 0.01$; *, $P_{corr} < 0.05$.
(TIFF)

**S11 Fig. Comparing classification performance with the full region–region feature space or the brain-wide mean FC for each SPI.** For each SPI, we compared case–control classification performance in each of the four disorders using either the full region–region pair input matrix (left) or the brain-wide average across all region–region pairs (right). Here, we show the distribution of all 100 test folds (10 repeats × 10 folds) as raincloud plots, where each dot represents one test fold. The solid horizontal line in each half-violin indicates the mean balanced accuracy and the dashed horizontal axis line denotes 50% balanced accuracy. Corrected resampled $T$-test results are indicated with ***, $P_{corr} < 0.001$.
(TIFF)

**S12 Fig. Raw covariance and normalized Pearson correlation coefficient show comparable case–control classification performance.** For each disorder (SCZ, BP, ADHD, and ASD), we plot the distribution of test balanced accuracy values across 100 folds (10 repeats × 10 folds) with the same `scikit-learn` pipeline for the raw covariance matrix across all region–pairs ('Cov', left) versus the normalized Pearson correlation matrix ('Pearson', right). The solid horizontal line in each half-violin indicates the mean balanced accuracy and the dashed horizontal axis line denotes 50% balanced accuracy.
(TIFF)

**S13 Fig. Cross-disorder classification analysis in the UCLA CNP cohort based on pairwise functional connectivity.** The out-of-sample balanced accuracy distribution (averaged across SCZ, BP, ADHD, and Control participants in the UCLA CNP cohort) is shown for each SPI on its own (**A**, $A_{FC}$) or with the inclusion of whole-brain local dynamics (**B**, $A_{FC\_combo}$). Each dot corresponds to one test fold (for a total of 100 data points, across 10 repeats × 10 folds), and the vertical black line in each half-violin corresponds to the mean balanced accuracy across test folds.
(TIFF)

**S14 Fig. Comparing classification performance with versus without inverse probability weighting. A**. For each disorder, the mean balanced accuracy per brain region is shown with no weighting ('None') or inverse probability weighting ('Balanced'). Lines correspond to each of 82 brain regions for SCZ, BP, and ADHD and each of 48 brain regions for ASD. Colors are included as a visual aid to highlight the difference in performance between the two weighting types, with red corresponding to higher balanced accuracy with inverse probability weighting and blue corresponding to lower balanced accuracy with inverse probability weighting. **B**. For

each disorder, the mean balanced accuracy per univariate time-series feature is shown with no weighting ('None') or inverse probability weighting ('Balanced'), as in A.
(TIFF)

**S15 Fig. Comparing normalization methods supports the use of the outlier-robust mixed sigmoid method.** Univariate time-series feature values were concatenated from all brain regions, with the resulting distributions depicted for all participants in the UCLA CNP cohort (**A**) or ABIDE cohort (**B**) with no normalization (upper row), $z$-score normalization (middle row), and outlier-robust mixed sigmoid normalization (bottom row; see Methods Sec. 'Case–control classification' for description). Pairwise SPI feature values were concatenated from all region–region pairs, with the resulting distributions depicted for all participants in the UCLA CNP cohort (**C**) or ABIDE cohort (**D**) with no normalization (upper row), $z$-score normalization (middle row), and outlier-robust mixed sigmoid normalization (bottom row).
(TIFF)

**S16 Fig. Null balanced accuracy metrics are approximately normally distributed across brain regions in all clinical cohorts.** The distribution of 1000 null balanced accuracy values (cross-validated over 10 repeats of 10 folds each) is shown for each brain region by disorder. Empirical null values are visualized as histograms, with the probability density overlaid as black lines for the normal distribution based on the mean and standard deviation for each null distribution. These normal distributions were used to compute $p$-values for each brain region based on the corresponding probability density function; the same procedure was also applied for univariate and pairwise time-series features. ASD brain regions are plotted separately from the SCZ, BP, and ADHD brain regions as different parcellation atlases were analysed for UCLA CNP (Desikan-Killiany atlas) and ABIDE cohorts (Harvard-Oxford cortical altas). The dashed vertical line marks 50% balanced accuracy in all plots.
(TIFF)

**S1 Table. The cross-validated classification results for A_region.** For each disorder, the mean balanced accuracy and standard deviation across the 100 test folds are indicated for each brain region, along with the raw and Benjamini–Hochberg corrected $p$-values.
(CSV)

**S2 Table. Results from an ordinary least squares linear regression of regional gray matter volume on diagnosis for each case–control comparison in the UCLA CNP cohort (i.e., SCZ, BP, and ADHD).** For each brain region and disorder, the estimated $\beta$ coefficient is indicated, along with the mean volume in each group and the raw and Benjamini–Hochberg corrected $p$-values.
(CSV)

**S3 Table. Results from case–control classification analysis using exclusively confound variables (age, sex, and/or mean FD) or confound variables plus rs-fMRI BOLD-derived time-series features.** For each confound type (e.g., "Age"), we present the mean and SD for balanced accuracy on its own or with each model per representation type (e.g., the left entorhinal cortex for A_region) per disorder.
(CSV)

**S4 Table. The cross-validated classification results for A_feature.** For each disorder, the mean balanced accuracy and standard deviation across the 100 test folds are indicated for each univariate time-series feature (measured across the entire brain), along with the raw and

Benjamini–Hochberg corrected $p$-values.
(CSV)

**S5 Table. The absolute Spearman rank correlation coefficient is indicated for each pair of univariate time-series features that significantly classified SCZ and/or ASD participants from controls.** For each pair of features, the correlation was estimated from the union of all brain regions from all SCZ, ASD, and control participants. The absolute value of the correlation coefficient is indicated as we focused on the magnitude of the relationship between each feature pair.
(CSV)

**S6 Table. The results from a Wilcoxon rank-sum test comparing the mean framewise displacement between each neuropsychiatric disorder group and the corresponding control participant group.**
(CSV)

**S7 Table. For each participant, the mean BOLD SD (averaged across all brain regions) is indicated, along with the mean framewise displacement.**
(CSV)

**S8 Table. The cross-validated classification results for $A_{uni\_combo}$.** For each disorder, the mean balanced accuracy and standard deviation across 100 test folds are indicated for the concatenated region × feature matrix, along with the raw and Benjamini–Hochberg corrected $p$-values.
(CSV)

**S9 Table.** Classifier type robustness analysis. For each disorder, we compared the cross-validated classification performance of each model (e.g., left entorhinal cortex in $A_{region}$) using the (1) standard linear SVM (as in our main analyses) along with (2) an additional L1-regularized SVM, (3) a radial basis function (RBF) SVM, (4) a random forest ensemble, and (5) a gradient-boosted ensemble classifier. The mean balanced accuracy is indicated for each of the five compared classifier types.
(CSV)

**S10 Table. Results from ASD vs. control classification analysis performed for all five representations ($A_{region}$ through $A_{FC\_combo}$) within each of the two largest sites in the ABIDE dataset: Site #20 ($N = 106$) and Site #5 ($N = 98$).**
(CSV)

**S11 Table. The cross-validated classification results for $A_{FC}$.** For each disorder, the mean balanced accuracy and standard deviation across the 100 test folds are indicated for each statistic for pairwise interactions (SPI, measured across all region–region pairs), along with the raw and Benjamini–Hochberg corrected $p$-values.
(CSV)

**S12 Table. The absolute Spearman rank correlation coefficient is indicated for each pair of SPIs that significantly classified SCZ and/or ASD participants from controls.** For each pair of SPIs, the correlation was estimated from the union of all region–region pairs from all SCZ, ASD, and control participants. The absolute value of the correlation coefficient is indicated as we focused on the magnitude of the relationship between each SPI pair.
(CSV)

**S13 Table. The cross-validated classification results for $A_{FC\_combo}$.** For each disorder, the mean balanced accuracy and standard deviation across the 100 test folds are indicated for each SPI (measured across all region–region pairs) combined with the full $A_{uni\_combo}$ matrix, along with the raw and Benjamini–Hochberg corrected *p*-values.
(CSV)

**S14 Table. The corrected resampled *T*-test results for the balanced accuracy based on each SPI with versus without the inclusion of whole-brain local dynamics ($A_{uni\_combo}$) in SCZ, BP, and ASD (the three groups for which at least one SPI significantly classified cases from controls).**
(CSV)

**S15 Table. The classification results from the one-shot model approach.** For each disorder, within each train/test split (out of 10 repeats of 10 train/test splits), the top-performing model was identified per representation type within the training fold. The top-performing model (identified within the training data) was then evaluated on the held-out test fold for the corresponding split, with this out-of-sample balanced accuracy reported. In the case of $A_{FC}$ and $A_{FC\_combo}$, the top-performing model was identified with an intermediate PCA within the training fold to reduce the dimensionality, though the out-of-sample performance was evaluated in the full feature space.
(CSV)

**S16 Table. Nested hyperparameter tuning for the regularization parameter in linear SVM.** For each disorder, we compared the cross-validated classification performance of each model (e.g., left entorhinal cortex in $A_{region}$) using either the fixed regularization parameter ($C = 1$) or by performing nested cross-validation within each held-out training set to optimize the $C$ parameter based on balanced accuracy. The mean balanced accuracy is indicated for each model type with versus without tuning the $C$ parameter.
(CSV)

**S17 Table. The first 25 principal components (PCs) derived from the region × feature matrix for each participant in both the UCLA CNP and ABIDE cohorts, corresponding to $A_{uni\_combo}$.**
(CSV)

## Acknowledgments

The authors thank Kieran Owens and Mac Shine for their feedback and suggestions for the manuscript and all participants and researchers involved in creating and openly sharing the fMRI datasets analyzed here. The authors also thank Patricia Tran and Preethom Pal for their contributions to related prior research projects.

## Author Contributions

**Conceptualization:** Alex Fornito, Ben D. Fulcher.

**Data curation:** Kevin Aquino, Linden Parkes.

**Formal analysis:** Annie G. Bryant.

**Methodology:** Annie G. Bryant, Kevin Aquino, Linden Parkes.

**Software:** Annie G. Bryant.

**Supervision:** Alex Fornito, Ben D. Fulcher.

**Visualization:** Annie G. Bryant.

**Writing – original draft:** Annie G. Bryant, Ben D. Fulcher.

**Writing – review & editing:** Kevin Aquino, Linden Parkes, Alex Fornito.

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
