## [Decision Letter · Decision Letter 0]

7 Sep 2024

Dear Ms Bryant,

Thank you very much for submitting your manuscript "Extracting interpretable signatures of whole-brain dynamics through systematic comparison" for consideration at PLOS Computational Biology.

As with all papers reviewed by the journal, your manuscript was reviewed by members of the editorial board and by several independent reviewers. In light of the reviews (below this email), we would like to invite the resubmission of a significantly-revised version that takes into account the reviewers' comments.

The reviewers bring up several points that need to be addressed in a revision, particularly R3's comments on model training and cross validation - either changes should be made to address their concerns or text modified to be clearer on what was actually done to train, validate and test the model. Further, other reviewers suggested changes to the text, adding other metrics left out in the original paper, and considering in-scanner motion in the classifiers.

We cannot make any decision about publication until we have seen the revised manuscript and your response to the reviewers' comments. Your revised manuscript is also likely to be sent to reviewers for further evaluation.

Sincerely,

Amy Kuceyeski

Academic Editor

PLOS Computational Biology

Hugues Berry

Section Editor

PLOS Computational Biology

The reviewers bring up several points that need to be addressed in a revision, particularly R3's comments on model training and cross validation - either changes should be made to address their concerns or text modified to be clearer on what was actually done to train, validate and test the model. Further, other reviewers suggested changes to the text, adding other metrics left out in the original paper, and considering in-scanner motion in the classifiers.

Reviewer's Responses to Questions

**Comments to the Authors:**

Reviewer #1: In the present manuscript Bryant et al. applied intra- and inert-regional measures of resting-state (rs) fMRI data to make case–control classifications of four neuropsychiatric disorders. The authors showed that, while simple intra-regional statistics performed fairly well, combining intra-regional properties with inter-regional couplings generally improved the classification performance.

I enjoyed reading the article, the work is clearly written, despite being very dense, and the methods are clearly presented. The results are interesting and consistent with previous findings and hypotheses. I think the proposed method will be very useful for other researchers studying large-scale brain activity, brain states, and neuropsychiatric disorders. Also, the method can be extended to include gene expression and neurotransmitter maps. Nevertheless, I have some concerns that the authors should address to improve the manuscript:

1- It has been shown that "glocal" measures of rs-fMRI are useful for discriminating between brain states. Such measures combine local and global properties of brain activity. For instance, it has been shown that the coupling between each brain region and the rest of the cortex provides an efficient statistic for classifying wakefulness versus anesthesia-induced loss of consciousness (Ponce-Alvarez et al., 2022; Tanabe et al., 2020). The performance of this local-global measure should be studied in the present work and compared to the results obtained by combining intra- and inter-regional measures.

Ponce-Alvarez A, Uhrig L, Deco N, Signorelli CM, Kringelbach ML, Jarraya B, Deco G (2022) Macroscopic quantities of collective brain activity during wakefulness and anesthesia. Cerebral Cortex 32(2), 298–311.

Tanabe S, Huang Z, Zhang J, Chen Y, Fogel S, Doyon J, Wu J, Xu J, Zhang J, Qin P et al. 2020. Altered global brain signal during physiologic, pharmacologic, and pathologic states of unconsciousness in humans and rats. Anesthesiology 132:1392–1406.

2- In the same vein, the performance of global measures should be tested and discussed. For example: a) the average correlation between all pairs of regions and b) the metastability measure (i.e., the standard deviation of the order parameter of signal phases) (Cabral et al., 2011). These global measures have been widely used to characterize different brain states.

Cabral J, Hugues E, Sporns O, Deco G. Role of local network oscillations in resting-state functional connectivity. Neuroimage. 2011;57:130–139.

3- Given that the SD and the Pearson correlation matrix have high classification performance, I think it would be interesting to test the performance of the covariance matrix.

Minor comments:

- In my opinion, the first two sentences of the introduction are too general and probably not necessary.

- Lines 260-261: “This supports the general use of linear time-series features like the fALFF to parsimoniously summarize relevant properties of rs-fMRI dynamics”. I disagree. Figure 2C shows that, for SCZ, the performance of fALFF ranks sixth, behind SD, DFA, periodicity, and mean. Additionally, for ASD, fALFF could not significantly distinguish cases from controls. Or am I missing something?

I hope the authors find these comments helpful.

Reviewer #2: Thank you for the opportunity to review the manuscript titled “Extracting interpretable signatures of whole-brain dynamics through systematic comparison” by Bryant and colleagues. The manuscript takes a data-driven approach to systematically compare various intra- and inter-regional dynamics from resting-state fMRI data in the human brain, with a specific focus on their applications in neuropsychiatric disorder classification. Specifically, the authors compare a relatively comprehensive and representative set of time-series and inter-regional connectivity measures (separately and combined) on the regional and whole brain level to benchmark their classification accuracy in 4 psychiatric disorders. They also perform various sensitivity analyses and compare different model selection procedures and classification algorithms. The findings show that linear time-series analysis techniques generally tend to perform well in the classification analysis. Interestingly, the results suggest that although certain measures might outperform others in specific disorders, combined intra- and inter-regional dynamics almost always improve the model performance, especially across 3 out of the 4 included disorders. This demonstrates that the included measures provide complimentary information about disorder-specific brain dynamics.

Altogether this manuscript is a novel study that builds on findings from recent studies, where intra- and inter-regional dynamics had been examined separately in the human brain. The study is rigorous and methodologically sound, taking a unique approach to systematically evaluate multiple functional data derivatives and their combinations, assessing their performance in psychiatric disorder classification. The manuscript is well-written and the work is novel, comprehensive, and thorough with interesting findings and compelling sensitivity analysis, providing an important contribution to the literature. I have a couple of minor comments that may help further improve the impact of the study and am happy to recommend a revised version of the manuscript for publication:

(1) The authors perform a thorough set of sensitivity analyses to ensure that the classification accuracy is independent from covariates and other potential factors including age, sex, and gray matter volume. They also assess whether participants’ in scanner motion is related to BOLD SD, where they identify a significant association between the two in ASD. Findings of these sensitivity analyses are compelling. However, I am wondering whether the authors have considered examining the potential influence of participants’ in scanner motion on performance accuracy directly (e.g., training a classifier with age, sex, and meanFD or including mean FD as a covariate in the analysis). It might be worthwhile further examining this given that the mean FD threshold used for data quality control is not necessarily a highly conservative threshold (as the authors acknowledge as well).

(2) Classification analysis is applied to each disorder separately. As the authors mention in the manuscript, data harmonization would be required to combine all the data because of differences in sample size and acquisition sites. However, I am wondering whether it would be possible to combine the UCLA sample for the data acquired in one of the two acquisition sites and assess the model performance, especially for the A_FC_combo analysis. This might be particularly interesting as it would test the classification accuracy of the included measures in data with multiple categories (rather than binary labels).

Reviewer #3: The paper is well-written and asks an interesting question on the predictive value of fMRI dynamics for classifying disease vs. control. Three types of disease are used, and wide variety of models are tested to answer the question comprehensively. Here are my main comments:

1. significance: it was not clear what is the state-of-the-art predictive performance for these disease classification problems, which features (univariate or FC) achieve that, and whether dynamic features are helping improve over that. Figure 3 shows Pearson FC has highest accuracy among the dynamic SPI measures.

It is also not clear if any scientific insight can be provided based on feature importances of the fitted models.

2. approach: I have serious concerns about potential overfitting in the results. The sample sizes are modest, the feature set is huge, and the authors are using linear SVM with the same default regularization (see lines 807-815 in page 27). Since the amount of regularization is not tuned to model size, it is possible that larger models with many features will be favored, which matches the empirical results. The discussion on variable selection is diffuse, and I would like to see the robustness of the findings when a state-of-the-art regularization ML method (LASSO, Ridge, even random forests or gradient boosting which are robust to default tuning parameter choices) is used.

3.Validation: Since there is no tuning parameter in the model, I could not understand how cross-validation (CV) is being used. In machine learning literature, it is customarty to reporst test or validation set errors, rather than CV-errors. It would be good to report out-of-sample performances on data that never touched by the model.

Finally, balanced accuracy measures depend on choice of decision threshold. So reporting area under ROC or precision-recall curve is perhaps a more fair metric when comparing models.

**Have the authors made all data and (if applicable) computational code underlying the findings in their manuscript fully available?**

Reviewer #1: Yes

Reviewer #2: Yes

Reviewer #3: Yes

PLOS authors have the option to publish the peer review history of their article (what does this mean?). If published, this will include your full peer review and any attached files.

Reviewer #1: **Yes: **Adrián Ponce-Alvarez

Reviewer #2: No

Reviewer #3: No
---

## [Decision Letter · Decision Letter 1]

3 Dec 2024

Dear Ms Bryant,

We are pleased to inform you that your manuscript 'Extracting interpretable signatures of whole-brain dynamics through systematic comparison' has been provisionally accepted for publication in PLOS Computational Biology.

Best regards,

Amy Kuceyeski

Academic Editor

PLOS Computational Biology

Hugues Berry

Section Editor

PLOS Computational Biology

Feilim Mac Gabhann

Editor-in-Chief

PLOS Computational Biology

Jason Papin

Editor-in-Chief

PLOS Computational Biology

Reviewer's Responses to Questions

**Comments to the Authors:**

Reviewer #1: The authors have addressed all my concerns. I congratulate them on their excellent work.

Reviewer #2: The authors have thoroughly addressed my comments and questions. The manuscript is substantially revised and improved. I have no further concerns.

Reviewer #3: The authors have addressed all the major concerns raised in my previous report.

**Have the authors made all data and (if applicable) computational code underlying the findings in their manuscript fully available?**

Reviewer #1: Yes

Reviewer #2: Yes

Reviewer #3: None

PLOS authors have the option to publish the peer review history of their article (what does this mean?). If published, this will include your full peer review and any attached files.

Reviewer #1: **Yes: **Adrián Ponce-Alvarez

Reviewer #2: No

Reviewer #3: No

---

## [Editor Report · Acceptance letter]

10 Dec 2024

PCOMPBIOL-D-24-01121R1 

Extracting interpretable signatures of whole-brain dynamics through systematic comparison

Dear Dr Bryant,

I am pleased to inform you that your manuscript has been formally accepted for publication in PLOS Computational Biology. Your manuscript is now with our production department and you will be notified of the publication date in due course.

With kind regards,

Dorothy Lannert
